# THE ROBUSTNESS-SECURITY PARADOX: CHANNEL-AWARE FEATURE LEARNING FOR ADVERSARIAL WATERMARK EXPLOITATION

## ABSTRACT

Watermarking is crucial for establishing provenance and detecting AI-generated content. While current approaches prioritize robustness against real-world distortions, we explore how the robustness-security tradeoff manifests in deep learning-based post-processing watermarks: robust watermarks necessarily increase the redundancy of detectable watermark patterns embedded in images, creating exploitable information leakage. Leveraging this insight, we introduce an attack framework that extracts watermark pattern leakage through multi-channel feature learning using pre-trained vision models. Unlike previous approaches that require extensive data or detector access, our method achieves both watermark removal (detection evasion) and watermark forgery attacks with just a single watermarked image in a no-box setting. Extensive experiments demonstrate our method outperforms state-of-the-art techniques by 52.9% in detection evasion rate and 9.6% in forgery accuracy, while preserving visual quality.

## 1 INTRODUCTION

Watermarking has become a critical safeguard for AI-generated content(AIGC) provenance and deepfake detection (Jiang et al., 2024; 2023), particularly as generative AI tools like Midjourney (Mid, 2024) and Sora (Sor, 2024) democratize AI access while amplifying misinformation risks (Kayleen Devlin, 2024). Major AIGC providers (OpenAI, Alphabet, Meta) are implementing imperceptible watermarking (Diane Bartz, 2024) to authenticate content origins, protect intellectual property, and mitigate reputational risks through traceable fingerprints.

Watermarking methods must exhibit robustness against various distortions to remain practical in real-world applications. This robustness, defined as the ability of the watermark to remain detectable despite modifications, is essential throughout the digital content lifecycle across platforms and channels. State-of-the-art watermarking frameworks address this challenge by incorporating distortion layers that simulate real-world perturbations during training, effectively "immunizing" systems against common distortions including lossy compression, noise addition, screen-capturing, and geometric transformations (e.g., cropping, scaling, rotation), etc. (Fang et al., 2022). These distortions often occur in combination, creating complex degradation patterns that naive approaches struggle to withstand. Distortion-aware training strategies significantly enhance watermark resilience throughout internet circulation, preserving the provenance information essential for content authentication systems (Tancik et al., 2020; Liu et al., 2019).

Despite advances in robustness against natural distortions, watermarking systems remain vulnerable to intentional adversarial attacks, including watermark removal (detection evasion) and watermark forgery (spoofing). These security vulnerabilities create significant societal and economic threats. In watermark removal attacks, adversaries eliminate embedded watermarks while preserving visual quality, enabling misappropriation of AI-generated imagery for unauthorized commercial use or distribution of Not-Safe-for-Work content without attribution. Conversely, forgery attacks involve transferring extracted watermark patterns to unwatermarked images depicting illegal or harmful content, allowing malicious actors to falsely attribute material to specific systems, making disinformation more credible or damaging organizational reputations. These attacks significantly challenge content authentication integrity.

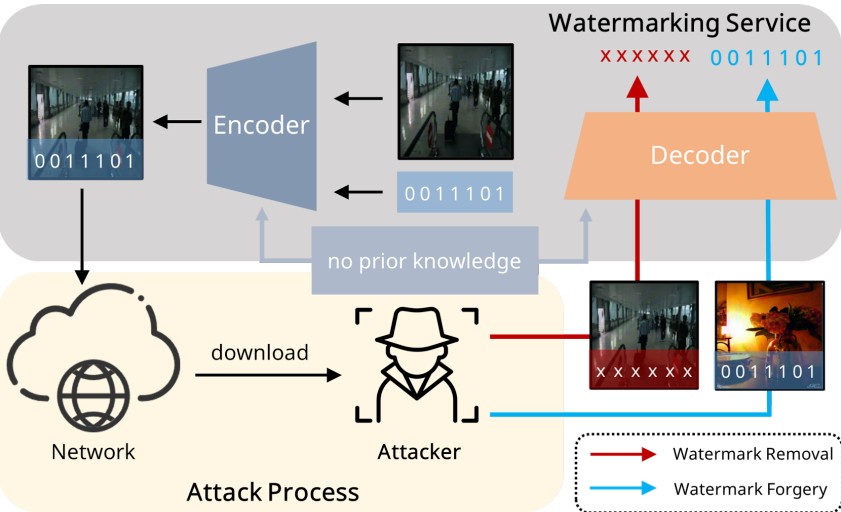

Figure 1: Demonstration of our attacks. An attacker can perform watermark detection evasion(watermark removal) and the watermark forgery with only one watermarked image without knowledge about the underlying watermarking systems. The attacker is free of copyright violation accusations as the extracted watermark is incorrect; the attacker can spread fake news by forging the watermark of an authoritative media.

Current watermark security research exhibits an asymmetric focus, with watermark removal dominating the literature, while studies on forgery attacks are increasing. *Watermark Removal*: We categorize relevant literature into adversarial example-based and reconstruction-based attacks. The former typically requires substantial watermark data for perturbation training (Lukas et al., 2024; Yang et al., 2024; Saberi et al., 2023) or direct detector access, sometimes even decoder knowledge (Lukas et al., 2024); the latter introduces noticeable modifications to the carrier image during reconstruction (Saberi et al., 2023; Zhao et al., 2023; Kassis & Hengartner, 2024). *Watermark Forgery*: This emerging research frontier has limited developed methodologies. Existing approaches rely on impractical assumptions (Kutter et al., 2000), require encoder access (Saberi et al., 2023), or demand extensive imagery collections with target watermarks, proving ineffective against schemes with dynamic elements like temporal signatures. In summary, current watermark removal methods are computationally expensive, based on unrealistic assumptions, or significantly alter image semantics, while forgery techniques remain impractical or resource-intensive, with both showing limited effectiveness against robust, distortion-aware watermarking systems.

In this paper, we reveal how the pursuit of robustness in watermarking systems creates a cascade of vulnerabilities. The well-established robustness-invisibility tradeoff forces watermarking systems to embed stronger, more redundant signals to survive distortions (De Vleeschouwer et al., 2002). However, we demonstrate that this reduction in invisibility has a critical security implication: the embedded patterns become learnable by neural networks, enabling both removal and forgery attacks. Thus, the robustness-invisibility tradeoff ultimately manifests as a robustness-security paradox—where distortion resistance directly facilitates adversarial exploitation. To systematically exploit this vulnerability, we **D**elve into the **A**spect of the **PA**radox **O**f Robust Watermarks (DAPAO) and propose the DAPAO attack, a unified attack framework that leverages multi-channel feature learning to extract watermark patterns from a single image (see Sec. 4.1 for details).

The DAPAO framework enables digital fingerprint extraction from watermarked images through **single-image analysis** in a strict **no-box setting** (zero system queries), achieving simultaneous watermark removal and clean-image forgery. To extract the watermark-related feature, we utilized typical neural networks to extract image features, represented in the form of multiple channels. We identify critical channels that have a bias toward watermark features and optimize learnable variables to align with the watermark characteristics and the semantics of the carrier image. Our method significantly outperforms state-of-the-art (SOTA) approaches in both watermark removal and forgery while preserving visual fidelity and semantic integrity.

**Summary of contributions.** In this paper, we make the following contributions:

Figure 2: Illustration of learning-based watermarking methods. $I$ and $wm$ are encoded into $I_{wm}$; the distortion layer produces $\hat{I}_{wm}$, which is decoded to $wm'$. Visual loss enforces imperceptibility, while MSE loss ensures robust extraction.

- We reveal the robustness-security paradox of post-processing watermark systems: Schemes improve watermark information redundancy to boost robustness against distortions, which results in watermark feature leakage that can be leveraged by attackers.
- Leveraging this observation, we propose DAPAO, a novel attack framework capable of both watermark removal and forgery against SOTA post-processing robust watermarking schemes. Our method requires only a single watermarked image for extraction and operates in a no-box setting. Code is available at https://anonymous.4open.science/r/dp-5971.
- Extensive experimental results demonstrate that our framework achieves notable improvement in attack performance over related work, with a 52.9% success rate improvement in watermark removal and a 9.6% improvement in forgery accuracy.

## 2 BACKGROUND

### 2.1 IMAGE WATERMARKING

Image watermarking includes injection, extraction, and verification. During watermark injection, an encoder $\mathcal{E}(\cdot, \cdot)$ receives the identification information $wm$ ("0011011" in Figure 2) and an original image $I$ as input and generates a watermarked image $I_{wm}$ with the key information embedded. During watermark extraction, the decoder $\mathcal{D}_{wm}(\cdot)$ extracts the identification key $wm'$ from the watermarked image $I_{wm}$ and then matches it with $wm$ to verify whether the target watermark exists. *Non-learning-based and Learning-based Watermarking:* non-learning-based methods build the encoder and decoder based on heuristics and signal processing techniques (Wang & buley, 2020). These traditional approaches typically operate in transform domains (e.g., DCT) and rely on hand-crafted rules for watermark insertion and detection. Learning-based methods deploy neural networks for the encoder and decoder, with parameters optimized with deep learning techniques. Empirical evidence demonstrates (Zhao et al., 2023) that learning-based methods exhibit more robustness against distortions. In particular, they can incorporate a distortion layer between the watermarked image and the decoder during the training phase (as shown in Figure 2). This architectural enhancement enables adversarial training by simulating potential real-world distortions that watermarked content might encounter during transmission or manipulation. The training objective is formulated to minimize the discrepancy between the decoder's output when processing a distorted watermarked image $\hat{I}_{wm}$ and ground truth watermark information $wm$. (Jiang et al., 2023). *Post-processing and In-processing Methods:* post-processing watermarking adds a watermark to an image post its generation, following the same process of watermarking a real image (Chopra et al., 2012; Al-Haj, 2007; Tancik et al., 2020). In contrast, in-processing watermarking embeds the identification message during the image generation process (Yu et al., 2021a;b; Wen et al., 2023).

### 2.2 WATERMARK REMOVAL AND WATERMARK FORGERY

Watermark removal means an attacker modifies a watermarked image to remove or disrupt the embedded watermark, causing the decoded bit string to deviate from the original identification information. Watermark forgery involves extracting the watermark information $wm$ from the watermarked

image $I_{wm}$ and embedding it into another non-watermarked image $I'$ to generate $I'_{wm}$, passing the verification of the watermark detector.

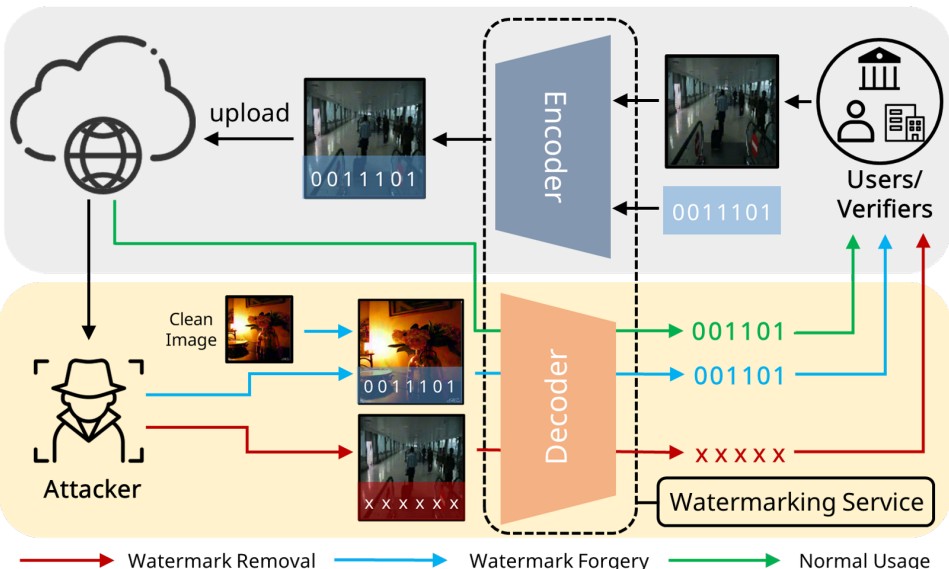

Figure 3: Typical watermarking application. Watermarks are embedded for copyright protection or content regulation, but attackers can use post-processing techniques to remove or forge the watermark.

## 3 PROBLEM FORMULATION

We primarily consider post-processing watermarking frameworks as they can be integrated with various generation models without modifying the underlying generative architecture, contributing to their widespread adoption in practical deployments. Figure 3 illustrates the use case of a typical watermarking system. The process consists of the stages of watermark injection (encoding), data circulation, and watermark extraction (decoding), as shown in the gray portion and the green data path of Figure 3. This process involves three key stakeholders, each with distinct roles and objectives. *Users/service providers*: Entities that generate or distribute content and apply watermarks to establish content provenance, protect intellectual property, or ensure accountability. *Verifiers*: Entities responsible for detecting and authenticating watermarks in circulated content. The verifier decodes the image to extract watermark information, thereby verifying content origins and integrity. *Attackers*: Adversarial actors who attempt to compromise the watermarking system through various attack vectors, including watermark removal or forgery, to circumvent content attribution or disseminate misleading information.

**Attacker's Goals and Capabilities**. Attackers have two types of objectives: *1) Watermark Removal(Detection Evasion):* The attacker's objective is to evade watermark detection by removing or disrupting embedded watermarks. The goal is to misappropriate intellectual property by removing creator attributions from watermarked content. *2) Watermark Forgery (False Attribution):* The attacker's objective is to transplant legitimate watermarks onto unauthorized content. The goal is to falsely attribute generated material to reputable sources or official entities. Attackers can download publicly available watermarked images attributable to the target entity. However, we assume three realistic limitations for the attackers: 1) Strict no-box constraints (no access to system internals or query capabilities for feedback); 2) Unavailability of the pre-watermarked (original) version of the target images, limiting comparative analysis capabilities; 3) The attacker must tackle sophisticated watermark methods integrating the distortion layer.

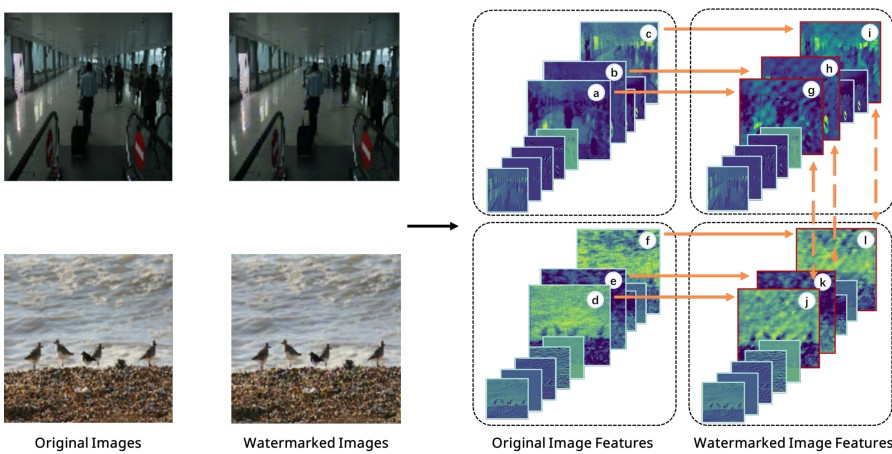

Figure 4: Feasibility study demonstration. Feature comparison between watermarked and original images (right part): the watermarking algorithm introduces specific patterns, shown by horizontal arrows (e.g., a–g, d–j); these patterns remain consistent across different image contents, as indicated by vertical arrows (e.g., g–j, h–k), while occupying only a small fraction of the overall features.

## 4 DAPAO ATTACKS

### 4.1 MOTIVATION AND CORE HYPOTHESIS

**Robustness-Driven Information Leakage**. Training post-processing watermarking models with increasingly diverse and severe distortions induces embedding redundancy. This redundancy is realized by relaxing imperceptibility constraints, leading to perceptible artifacts in the watermarked image. Although previously reported in early heuristic approaches (De Vleeschouwer et al., 2002), We extend the analysis to modern deep learning frameworks with distortion layers and hypothesize that these artifacts, though subtle, correlate with the watermark signal and constitute information leakage, serving as an unintended side channel that reveals watermark patterns.

**Capturing Information Leakage**. To test this hypothesis, we employ feature extractors to analyze potential robustness-induced patterns. In our feasibility study, we embed watermarks into multiple images using the same robust watermarking algorithm (PIMoG (Fang et al., 2022)), feed the watermarked images into a feature extractor, and then visualize the extracted representations. (Figure 4). We observe that: 1) The extracted features successfully capture leaked watermark patterns that are not easily perceptible through visual inspection of the watermarked images(The horizontal arrows in Figure 4, e.g., a–g); 2) This leakage is concentrated in specific feature channels rather than uniformly distributed; 3) These patterns demonstrate consistency across diverse image content—their invariance across different images confirms they encode the shared watermark signal(Vertical arrows in Figure 4, e.g., g–j). These findings reveal a fundamental paradox: mechanisms designed to enhance robustness against distortions inadvertently create security vulnerabilities, enabling adversaries to extract and exploit watermark patterns for removal or forgery.

**Localization and Exploitation of Leakage Features**. Building on these validated observations, effective watermark attacks must overcome two challenges: 1) automatically identifying features that encode leakage, and 2) exploiting these features to execute the attack. According to Sec. 4.1, leakage features occupy only a small fraction of the feature space and can therefore be modeled as statistical outliers. This property allows us to employ outlier-sensitive clustering algorithms to automatically localize them. Building on this, and inspired by adversarial example optimization, we design a learnable perturbation that approximates the leaked watermark information. The optimization process is task-specific, enabling either watermark removal or forgery.

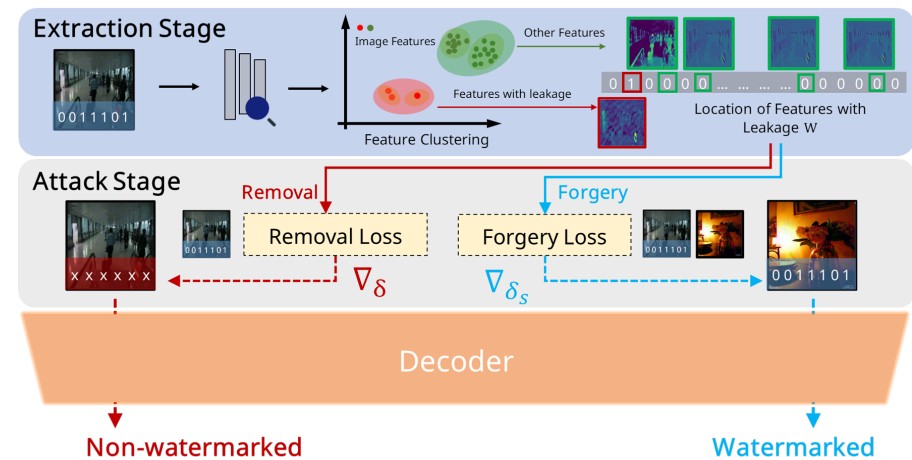

Figure 5: An overview of our attack. During the extraction stage, features from watermarked images are extracted using a feature extraction network and clustered to identify potential watermark leakage. These features are then used as prior knowledge to guide the optimization of perturbations for evasion and forgery.

## 4.2 REMOVAL AND FORGERY ATTACKS

Building on the above observations, we introduce two attack strategies that exploit robustness-induced leakage: watermark removal and watermark forgery. The overall framework, illustrated in Figure 5, is organized into three parts, which are discussed below.

**Extracting Leakage Features**. As shown in the *Extraction Stage* of Figure 5, suppose we have a watermarked image $I_{wm}$ embedded with an unknown watermark $wm$. Guided by Sec. 4.1, we first extract multi-channel features $\mathcal{F}(I_{wm})$ using a feature extractor $\mathcal{F}(\cdot)$. We then apply K-Means clustering to partition the feature channels into $k$ clusters and select the $n$ clusters with the fewest samples, thereby reducing the inclusion of irrelevant features that may dilute the leakage signal. We denote their channel indices as $\mathcal{W}$. The extracted leakage features are thus represented as $\mathcal{W} \cdot \mathcal{F}(I_{wm})$, which capture the watermark-correlated patterns identified through clustering.

**Watermark Removal**. As shown in the *Attack Stage* of Figure 5, having identified the leakage features after the leakage-feature extraction, we introduce a perturbation $\delta$ that disrupts the leaked information while preserving visual quality. This is formulated as(i.e., the *removal loss* in Figure 5):

$$\min_\delta -\mathcal{L}(\mathcal{W} \cdot \mathcal{F}(I_{wm}), \mathcal{W} \cdot \mathcal{F}(I_{wm} + \delta)), \text{s.t.} ||\delta||_\infty < \epsilon_1 \tag{1}$$

where $\mathcal{L}(\cdot, \cdot)$ is the loss function that measures the distance between two features, and $\epsilon_1$ is a perturbation budget. We use Projected Gradient Descent (PGD) (Mądry et al., 2017) to solve the optimization problem in Eq 1. The optimized perturbation $\delta$ effectively masks the watermark patterns by maximizing the dissimilarity between the original and perturbed leakage features. Finally, we complete the attack through $I_{wm} + \delta$. Our detailed algorithm is shown as Algorithm 1 in Appendix A.4.

**Watermark Forgery**. As illustrated in the *Attack Stage* of Figure 5, we repeat the leakage-feature extraction to obtain $\mathcal{W}$ and optimize Eq. 1 to learn $\delta$, referred to as Stage I. This learned $\delta$ approximates the watermark pattern extracted from $I_{wm}$. However, the learned $\delta$ alone cannot fulfill the forgery purpose as the watermarking model takes into account the alignment between the watermark information and the semantic content of the carrier image during the decoding process (See Table. 13 in Appendix. A.3.6). After Stage I, an additional optimization term is introduced to learn another perturbation, $\delta_s$, that captures the corresponding semantic information to meet the alignment requirement. This can be formulated as(i.e., the *forgery loss* in Figure 5):

$$\min_{\delta_s} \mathcal{L}((1 - \mathcal{W}) \cdot \mathcal{F}(I_{wm} + \text{sg}(\delta)), (1 - \mathcal{W}) \cdot \mathcal{F}(I' + \delta_s)), \text{s.t.} ||\delta_s||_\infty < \epsilon_2 \tag{2}$$

Where $I'$ denotes the clean image and $\text{sg}(\cdot)$ denotes the stop-gradient operation. This process is referred to as Stage II. We use PGD (Mądry et al., 2017) to solve Eq. 1 and Eq. 2. The forged image is obtained as $I' - \delta$ or $I' - \delta + \delta_s$. The complete algorithm is given in Algorithm 2 (Appendix A.4).

## 5 EVALUATION

### 5.1 SETUP

We evaluate our method on COCO (Lin et al., 2014a) and DIV2K (Agustsson & Timofte, 2017) datasets, sampling 100 images per dataset embedded with diverse watermarking algorithms for evasion/forgery testing. For cross-image forgery attacks, we construct 100 image pairs using non-overlapping COCO images randomly coupled with watermarked counterparts.

**Watermark Setting**. To evaluate our attack method, we test five publicly available post-processing watermarking methods. These cover a diverse range of approaches, including those resilient to physical-world distortions, as well as the watermark adopted in Stable Diffusion (Rombach et al., 2022): RivaGAN (Zhang et al., 2019), StegaStamp (Tancik et al., 2020), HiDDeN (Zhu et al., 2018), PIMoG (Fang et al., 2022), and CIN (Ma et al., 2022), with corresponding watermark capacities of 32 bits, 100 bits, 30 bits, 30 bits, and 30 bits, respectively. Regarding the detection thresholds, we uniformly set them to {0.95, 0.9, 0.85, 0.8, 0.75, 0.7, 0.65, 0.6, 0.55}, and also establish a threshold calculated based on watermark length $n$: rejecting the null hypothesis with $p < 0.05$. The corresponding thresholds are calculated to be 0.625, 0.57, 0.6, 0.6 and 0.6.

**Attack Benchmarking**. For removal attacks, we compared our approach with traditional image degradation techniques such as JPEG compression with quality 25, Gaussian noise with standard deviation 0.1, and Gaussian blur with kernel size 5 and standard deviation 2. These common image processing operations could potentially interfere with watermarks (Zhao et al., 2023; Jiang et al., 2023). We also benchmark with related attack methods, including WmAttacker (Zhao et al., 2023) and WmRobust (Saberi et al., 2023). For forgery attacks, we compare our method with existing forgery techniques, including CopyAttack (Kutter et al., 2000), Steganalysis (Yang et al., 2024), and WmRobust (Saberi et al., 2023).

**Evaluation Metrics**. Visual fidelity is evaluated using SSIM (Wang et al., 2004) and Peak Signal-to-Noise Ratio (PSNR) between attacked and original watermarked images. Attack effectiveness is measured by bit accuracy and success rate (SR). Bit accuracy is the proportion of correctly matched bits between the extracted watermark and the ground truth. SR is the fraction of successfully attacked samples, defined as bit accuracy falling below a threshold for evasion attacks and above the threshold for forgery attacks. The detection threshold corresponding to a False Positive Rate (FPR) of 0.05 is determined according to the default configuration of each watermarking method.

**Parameter Settings**. We evaluate different parameter configurations through extensive experiments (detailed in Appendix A.2). Based on parameter tuning, we identified K-Means clustering with $k = 4, n = 2$ as achieving the best balance between attack effectiveness and visual quality. We used SSIM loss in the subsequent experiments. Our experiments showed that SSIM loss consistently outperforms L1/L2 losses across different watermarking methods. We employ DenseNet (Huang et al., 2017) pretrained on ImageNet as the feature extraction module $\mathcal{F}(\cdot)$, with experiments confirming similar performance using ResNet, demonstrating robustness to backbone architecture choice. We use AdamW (Loshchilov, 2017) as the optimizer, with a learning rate of 0.15 and a weight decay of 1e-4. $\epsilon_1$ is 0.08 for evasion. $\epsilon_1$ is 0.08 and $\epsilon_2$ is 0.018 for watermark forgery. All experiments are conducted on an NVIDIA GeForce RTX 3090 GPU.

Table 1: Overall watermark removal performance on the COCO dataset. Detailed results of full threshold candidates are shown in Appendix A.3.2.

| Methods | PIMoG(th=0.6) | | | HiDDeN(th=0.6) | | | StegaStamp(th=0.57) | | | RivaGAN(th=0.625) | | | CIN(th=0.6) | | |
|---|---|---|---|---|---|---|---|---|---|---|---|---|---|---|---|
| | SR↑ | SSIM↑ | PSNR↑ | SR↑ | SSIM↑ | PSNR↑ | SR↑ | SSIM↑ | PSNR↑ | SR↑ | SSIM↑ | PSNR↑ | SR↑ | SSIM↑ | PSNR↑ |
| WmAttacker | 0 | 0.749 | 28.503 | 0.32 | 0.628 | 25.026 | 0 | 0.715 | 27.586 | 0.13 | 0.601 | 24.555 | 0.01 | 0.614 | 28.062 |
| WmRobust | 0.26 | 0.829 | 35.064 | 0.34 | 0.808 | 35.363 | 0.6 | 0.808 | 33.75 | 0.47 | 0.842 | 35.648 | 0.66 | 0.77 | 33.595 |
| JPEG | 0 | **0.897** | **37.538** | 0.36 | 0.813 | 34.161 | 0 | 0.867 | 35.976 | 0.15 | 0.793 | 33.289 | 0.02 | **0.898** | **39.238** |
| Gaussian | 0.01 | 0.197 | 26.532 | **0.81** | 0.406 | 26.441 | 0 | 0.365 | 26.496 | 0.01 | 0.157 | 26.579 | 0 | 0.324 | 26.447 |
| GaussianBlur | 0 | 0.771 | 31.362 | 0.05 | 0.651 | 30.567 | 0 | 0.713 | 30.123 | 0 | 0.615 | 28.433 | 0 | 0.761 | 31.495 |
| **Ours** | **0.87** | 0.876 | 36.403 | 0.77 | **0.889** | **36.644** | **1** | **0.895** | **36.703** | **1** | **0.872** | 35.851 | **0.78** | 0.809 | 34.801 |

Table 2: Overall forgery performance on the COCO dataset. Detailed results of full threshold candidates are shown in Appendix A.3.3.

| Methods | PIMoG(th=0.6) | | | HiDDeN(th=0.6) | | | StegaStamp(th=0.57) | | | RivaGAN(th=0.625) | | | CIN(th=0.6) | | |
|---|---|---|---|---|---|---|---|---|---|---|---|---|---|---|---|
| | SR↑ | SSIM↑ | PSNR↑ | SR↑ | SSIM↑ | PSNR↑ | SR↑ | SSIM↑ | PSNR↑ | SR↑ | SSIM↑ | PSNR↑ | SR↑ | SSIM↑ | PSNR↑ |
| CopyAttack | 0.21 | 0.726 | 20.58 | 0.25 | 0.778 | 21.739 | 0.11 | 0.747 | 20.831 | 0.09 | 0.704 | 22.19 | 0.16 | 0.736 | 20.014 |
| Steganalysis | 0.18 | 0.907 | **34.524** | 0.08 | **0.902** | 34.451 | 0.11 | **0.918** | **34.42** | 0.13 | **0.902** | 34.483 | 0.3 | **0.912** | **34.572** |
| WmRobust | 0.89 | **0.915** | 31.02 | 0.68 | 0.736 | 26.035 | 0.92 | 0.832 | 29.747 | 0.12 | 0.78 | 31.841 | 1 | 0.818 | 28.471 |
| **Ours** | **1** | 0.809 | 33.485 | **0.86** | 0.85 | **37.144** | **1** | 0.827 | 34.01 | **0.24** | 0.774 | **34.795** | **1** | 0.807 | 33.716 |

## 5.2 RESULTS AND ANALYSIS

In this section, we present the detailed results of our attacks and provide an analysis of the relevant findings. More detailed experimental results can be found in Appendix A.3.

**Watermark Removal**. Tables 1 and 12 demonstrate the superiority of our method across diverse watermarking algorithms and datasets. While traditional degradation methods (e.g., JPEG) preserve high visual quality, they fail to effectively remove watermarks, achieving near-zero success rates in most cases. Among methods with meaningful attack capability, our approach achieves the highest success rates, with an average of 88% on both COCO and DIV2K, while maintaining competitive visual fidelity (SSIM > 0.8 in all cases). Specifically, our method outperforms the closest competitor, WmRobust, by 41.8% and 64% in average success rate on COCO and DIV2K, respectively, with comparable or better visual quality. Figure 38 shows qualitative examples of our evasion attacks.

**Watermark Forgery**. Table 2 reports forgery attack results. Our method achieves consistently high success rates in a strict no-box setting without system access, improving on the second-best method by an average of 9.6% while preserving high visual quality. By contrast, CopyAttack and Steganalysis remain critically low (SR < 0.3), and WmRobust, though effective on some watermarks (e.g., SR = 1 on CIN), relies on privileged encoder access that is impractical in real attacks. Figure 39 shows successful forgeries. That said, we note a limitation: for RivaGAN, while our removal attack is highly effective, forgery rates are lower due to its tighter coupling between watermark and semantics, which restricts transferring leaked information to arbitrary clean images (see Sec. 5.3).

**Remarks**. Overall, our method reliably uncovers and exploits the leaked watermark information across representative post-processing watermark algorithms, achieving strong watermark removal ■, while the forgery attack effect depends on the degree of watermark and image semantics alignment. The results reveal severe vulnerabilities in well-recognized schemes and provide insights for future post-processing robust watermark design.

Table 3: Ablation study for watermark removal. The experiments are divided into two parts: feature extraction network and feature channel position retrieval. **w/o** $\mathcal{F}$ indicates the use of the original image without feature extraction, while **w/o** $\mathcal{W}$ indicates the use of all channels without selective retrieval. Detailed results of full threshold candidates are shown in Appendix A.3.4.

| Methods | PIMoG(th=0.6) | | | HiDDeN(th=0.6) | | | StegaStamp(th=0.57) | | | RivaGAN(th=0.625) | | | CIN(th=0.6) | | |
|---|---|---|---|---|---|---|---|---|---|---|---|---|---|---|---|
| | SR↑ | SSIM↑ | PSNR↑ | SR↑ | SSIM↑ | PSNR↑ | SR↑ | SSIM↑ | PSNR↑ | SR↑ | SSIM↑ | PSNR↑ | SR↑ | SSIM↑ | PSNR↑ |
| $\mathcal{W}$, $\mathcal{F}$ | 0.87 | **0.876** | 36.403 | 0.77 | **0.889** | 36.644 | 1 | **0.895** | 36.703 | 1 | **0.872** | 35.851 | 0.78 | **0.809** | 34.801 |
| w/o $\mathcal{W}$, $\mathcal{F}$ | 1 | 0.331 | 28.591 | **0.86** | 0.324 | 27.93 | 1 | 0.345 | 28.41 | 1 | 0.373 | 27.829 | 0.89 | 0.314 | 28.789 |
| w/o $\mathcal{W}$, w/o $\mathcal{F}$ | 0 | 0.184 | 27.605 | 0.52 | 0.163 | 27.644 | 0 | 0.195 | 27.615 | 0.3 | 0.118 | 27.973 | 1 | 0.142 | 27.762 |

Table 4: Ablation study for watermark forgery. The results are divided into three parts: Stage I, Only Stage II and Stage I + Stage II. Detailed results of full threshold candidates are shown in Appendix A.3.5.

| Methods | PIMoG(th=0.6) | | | HiDDeN(th=0.6) | | | StegaStamp(th=0.57) | | | RivaGAN(th=0.625) | | | CIN(th=0.6) | | |
|---|---|---|---|---|---|---|---|---|---|---|---|---|---|---|---|
| | SR↑ | SSIM↑ | PSNR↑ | SR↑ | SSIM↑ | PSNR↑ | SR↑ | SSIM↑ | PSNR↑ | SR↑ | SSIM↑ | PSNR↑ | SR↑ | SSIM↑ | PSNR↑ |
| Stage I | 1 | **0.809** | 33.485 | 0.29 | **0.887** | 35.466 | 1 | **0.827** | 34.01 | 0.21 | **0.898** | 35.495 | 1 | **0.807** | 33.716 |
| Only Stage II | 1 | 0.702 | 32.194 | 0.56 | 0.588 | 31.294 | 0.98 | 0.701 | 32.172 | 0.19 | 0.613 | 31.29 | 1 | 0.701 | 32.151 |
| Stage I + Stage II | 1 | 0.684 | 31.818 | **0.86** | 0.85 | **37.144** | 0.99 | 0.685 | 31.779 | **0.24** | 0.774 | 34.795 | 1 | 0.679 | 31.769 |

## 5.3 ABLATION STUDY AND TRADE-OFF ANALYSIS

**Watermark Removal**. Table 3 reports the results of our evasion attack and the results of ablating different parts of the method, validating the effectiveness of our design. Using only $\mathcal{F}$ without

channel selection achieves strong attacks but severely degrades image quality. Optimizing directly on the original image worsens visual perception and reduces attack effectiveness. These results confirm that watermark leakage can be captured via feature extraction, and precise localization can minimize image distortion.

**Watermark Forgery**. Table 4 presents the ablation results of our forgery attack. Note that Only Stage II is shown as follows: $\min_{\delta_s} \mathcal{L}((1 - \mathcal{W}) \cdot \mathcal{F}(I_{wm}), (1 - \mathcal{W}) \cdot \mathcal{F}(I' + \delta_s)), s.t.||\delta_s||_\infty < \epsilon$, where $\epsilon$ is 0.078. Note that generating the perturbation $\delta$ through Stage I is unnecessary at this point. Experimental results show that we successfully forge watermarks of three algorithms—PIMoG, StegaStamp, and CIN, achieving optimal performance using Stage I, where the attack merely extracts leaked watermark information and embeds it into a clean image. This indicates that these schemes do not enforce a strong alignment between watermark information and image content. However, for RivaGAN and HiDDeN, the Stage I forgery attack performs poorly. Combining Stage I and Stage II obviously improves the forgery effect, though a slight degradation of visual quality is observed. This suggests that RivaGAN and HiDDeN facilitate a more substantial alignment between image semantics and watermark information.

**Robustness-Security Trade-off**. We designed an adaptive defense against our attack method (see Appendix A.1.2). The experiments (Table 6) show that mitigating leakage reduces the attack success rate by 18%, but it also compromises the watermark's robustness against distortions, with the JPEG interference success rate increasing by 24%, supporting our trade-off hypothesis.

## 6 RELATED WORK

### 6.1 WATERMARK REMOVAL

*Destruction and Reconstruction*: the watermarked image firstly undergoes a certain level of degradation, followed by reconstruction to obtain a purified image. The mainstream approach for this method involves adding noise to the image and then using generative models, such as Diffusion Models (DM) (Ho et al., 2020), for reconstruction and generation (An et al., 2024; Saberi et al., 2023; Zhao et al., 2023). In contrast, UnMarker (Kassis & Hengartner, 2024) employs a learnable filter to process the watermarked image, supplemented by visual loss functions to ensure and enhance the visual quality of the attack results. *Adversarial Attacks*: transferring classic adversarial attack methods to the watermarking domain primarily targets the decoder of watermark models. WEVADE (Jiang et al., 2023) incorporates both black-box and white-box adversarial attack methods. Lukas et al. (Lukas et al., 2024) employ a surrogate model closely resembling the target model to perform transfer attacks. WmRobust (Saberi et al., 2023) requires a dataset containing both watermarked and non-watermarked images to train a feature classifier subjected to adversarial attacks. The attacks are transferred to the target watermark detection module. Similarly, WAVES (An et al., 2024) relies on a relevant watermark dataset for surrogate attacks but introduces a more detailed classification of watermark data. Hu et al. (Hu et al., 2024) explore the feasibility of large-scale ensemble surrogate models for transfer attacks against target watermark models.

### 6.2 WATERMARK FORGERY

CopyAttack (Kutter et al., 2000) was the first to introduce the concept of spoof attacks and proposed a method for predicting watermarks in unknown watermarking algorithm scenarios, embedding them into other images to achieve forgery. WmRobust (Saberi et al., 2023) proposes an attack leveraging the encoder of the watermark model to embed noise with a watermark and applies fine-tuning to generate forged watermarks. Steganalysis (Yang et al., 2024) computes a residual by statistically analyzing a dataset of watermarked images and unpaired clean images. This residual is then used to facilitate watermark forgery.

## 7 CONCLUSION

We propose DAPAO attack, a framework that requires only one image for watermark extraction, effectively achieving both detection evasion and watermark forgery against cutting-edge post-processing robust watermarking methods. Our work empirically demonstrates how the robust-

ness–invisibility trade-off in modern deep post-processing watermarking systems can lead to security vulnerabilities. Our attack reaches an average success rate of 88% in detection evasion (about 52.9% higher than existing evasion attacks) and an average success rate of 82% in forgery (approximately 9.6% higher than current forgery studies). To address these vulnerabilities, mitigating DAPAO requires improved schemes to minimize information leakage.We discuss potential defense methods in Appendix A.1.

## 8 REPRODUCIBILTY STATEMENT

We have taken several steps to ensure the reproducibility of our work. The main paper provides a complete description of the proposed methodology, including the formulation of the leakage-driven attack framework (Sec. 4.1) and the optimization objectives for both evasion and forgery attacks . Additional implementation details, hyperparameter choices, and ablation studies on clustering parameters, loss functions, and feature extractors are included in the Sec. 5 and Appendix A.2. We also provide pseudocode for the main algorithms in Appendix A.4. To facilitate replication, we release our core code and scripts for reproducing at an anonymous repository: `https://anonymous.4open.science/r/dp-5971`.

## 9 ETHICS STATEMENT

Watermarking is an avenue for AIGC provenance and detection, preventing potential misbehavior such as the spread of misinformation, copyright violation, and adversarial false attribution. Our work primarily underscores novel threats to modern learning-based watermarking schemes prioritizing robustness against real-world distortions. In theory, attackers could exploit these vulnerabilities to compromise watermarks, potentially harming users and service providers. However, the watermarking methods analyzed in this study are all open-source and research-focused, while the real-world deployment of invisible and robust watermarks remains in its early stages. Therefore, we believe making our work public has no direct negative impact. Conversely, we believe our findings have a positive societal impact by exposing a fundamental vulnerability in existing robust watermarking techniques, thereby preventing potential covert exploitation by adversaries and offering valuable insights for developing more secure image watermarking solutions.

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

# A  APPENDIX

## A.1  DISCUSSION AND LIMITATIONS

### A.1.1  ATTACK EFFECTIVENESS AGAINST IN-PROCESSING WATERMARKS

In-processing watermarks usually demonstrate a stronger coupling effect with the semantics of imagery, probably presenting insufficient pattern leakage for our method to utilize. For example, Tree-Ring (Wen et al., 2023) subtly influences the entire image generation process by embedding a pattern structured in the Fourier domain into the noise vector for sampling. This watermarking significantly connects with the image content, and our attacks exhibit limited performance on it.

Table 5: 100 prompts were randomly selected from the COCO dataset, one watermarked image was generated for each using Stable Diffusion, and our evasion method was then applied.

| Watermark Method | SR↑ | PSNR↑ | SSIM↑ |
|---|---|---|---|
| Tree-Ring (FPR=$10^{-3}$) | 0.12 | 33.177 | 0.690 |

### A.1.2  DEFENSES AGAINST DAPAO

We urgently need to develop defenses for post-processing watermarks against our attacks, considering these methods have wider application scenarios.

**Adaptive Defense**. One promising direction is to enhance adversarial training (Sec. 2) by incorporating leakage estimation, thereby balancing robustness and security. To this end, we implemented an adaptive defense that guides the leakage-related features of watermarked images to align with the semantic features of the original images during training. As a control, we trained another model with identical settings but without the adaptive component, and then applied our evasion attack algorithm. We subsequently performed removal attacks—using both our proposed method and JPEG compression—on models with and without this adaptive design. The differences in attack success rates across the two settings provide further evidence supporting the validity of our assumption. The results are summarized in the table below:

Table 6: Adaptive design.

| Attack | Target | Avg. Bit ACC↑ |
|---|---|---|
| JPEG | HiDDeN | 0.36 |
| Ours | HiDDeN | 0.78 |
| JPEG | HiDDeN(adaptive design) | 0.6 |
| Ours | HiDDeN(adaptive design) | 0.6 |

We observed that incorporating adaptive design into the HiDDeN model indeed improved its resistance to our proposed attack: the attack success rate dropped from 0.78 to 0.60, indicating an 18% reduction compared to the model without adaptive design. However, the same model became more vulnerable to JPEG compression, with the interference success rate increasing from 0.36 to 0.60—an increase of 24%.

We believe this result supports the intuition that mitigating the leakage-related features can provide partial defense against our method. This not only aligns with the idea of adaptive defense, but also further suggests that the effectiveness of our attack stems from the presence of such leakage. At the same time, the reduction in robustness against distortions highlights the inherent trade-off, which supports our hypothesis.

**Tighter Coupling with Semantic Features**. As discussed in Appendix. A.1.1 and inspired by the spoofing experiments Table. 4, we suggest that stronger semantic coupling between the watermark and image content may enhance watermark security. These findings point to in-processing

watermarking approaches as a promising direction for future research but the application scope of in-process watermarking needs to be further expanded.

However, this watermarking approach has a limited range of applications. It necessitates significant modifications to existing image synthesis algorithms and embedding information into non-synthetic images requires additional design efforts, thereby reducing scalability.

## A.2    ABLATION AND ANALYSIS

**Effect of Clustering Parameters** $k$ **and** $n$. We explored a reasonable range of k and the number of selected smallest clusters n and empirically chose the final setting based on validation. We ablated k and n on PIMoG and HiDDeN.

Table 7: Ablation study on k and n for evasion attacks on PIMoG

Table 8: Ablation study on k and n for evasion attacks on HiDDeN

| (k,n) | SR↑ | PSNR↑ | SSIM↑ |
|---|---|---|---|
| (2,1) | 0.672 | 36.563 | 0.901 |
| (3,1) | 0.627 | 36.619 | 0.9 |
| (3,2) | 1 | 32.885 | 0.741 |
| (4,1) | 0.72 | 36.763 | 0.9 |
| (4,2) | 0.87 | 35 | 0.768 |
| (4,3) | 0.987 | 32.396 | 0.687 |
| (5,1) | 0.667 | 36.341 | 0.896 |
| (5,2) | 0.969 | 33.972 | 0.829 |
| (5,3) | 1 | 32.124 | 0.739 |
| (6,1) | 0.687 | 36.522 | 0.898 |
| (6,2) | 1 | 34.259 | 0.84 |
| (6,3) | 1 | 32.728 | 0.79 |

| (k,n) | SR↑ | PSNR↑ | SSIM↑ |
|---|---|---|---|
| (2,1) | 0.50 | 37.529 | 0.910 |
| (3,1) | 0.50 | 37.563 | 0.911 |
| (3,2) | 0.81 | 32.203 | 0.725 |
| (4,1) | 0.679 | 33.376 | 0.905 |
| (4,2) | 0.77 | 36.644 | 0.889 |
| (4,3) | 0.83 | 31.758 | 0.675 |
| (5,1) | 0.44 | 37.530 | 0.907 |
| (5,2) | 0.78 | 34.323 | 0.847 |
| (5,3) | 0.70 | 32.461 | 0.764 |
| (6,1) | 0.46 | 37.481 | 0.910 |
| (6,2) | 0.74 | 34.746 | 0.858 |
| (6,3) | 0.74 | 33.002 | 0.808 |

Results show that, for a fixed $k$, increasing $n$ raises the success rate but progressively sacrifices visual quality, as it introduces more unrelated features. This trend is consistent with Table 3 in the main paper: using more features improves success rate but severely degrades visual quality. Conversely, using $n = 1$ yields limited effectiveness because too few leakage-related features are selected. Increasing k does not consistently improve success rates and introduces fluctuations in visual quality.

**Effect of Loss Function**. We also performed additional experiments using L1 and L2 loss for watermark removal on PIMoG and HiDDeN. As shown in Table 9, using SSIM consistently improved attack success rates compared to L1/L2, validating this design choice.

Table 9: Ablation study comparing different loss function choices.

| Target | Loss | SR↑ | PSNR↑ | SSIM↑ |
|---|---|---|---|---|
| PIMoG | L1 | 0.596 | 33.351 | 0.866 |
| PIMoG | L2 | 0.680 | 31.664 | 0.847 |
| PIMoG | SSIM | 0.870 | 36.403 | 0.876 |
| HiDDeN | L1 | 0.020 | 36.879 | 0.885 |
| HiDDeN | L2 | 0.170 | 33.629 | 0.839 |
| HiDDeN | SSIM | 0.770 | 36.644 | 0.889 |

**Effect of Feature Extractor**. We replaced DenseNet with ResNet as the feature extractor and conducted watermark removal and forgery experiments on two watermarking algorithms. As shown in Table 10. and Table 11. below, the results remained consistent with those obtained using DenseNet, demonstrating that our method does not rely on a specific backbone and that the underlying assumption still holds.

Table 10: Evasion attack using ResNet as the feature extraction network

Table 11: Spoof attack using ResNet as the feature extraction network

| Target | SR↑ | PSNR↑ | SSIM↑ |
|--------|------|--------|--------|
| PIMoG | 1.00 | 33.271 | 0.739 |
| HiDDeN | 0.90 | 33.138 | 0.745 |

| Target | SR↑ | PSNR↑ | SSIM↑ |
|--------|------|--------|--------|
| PIMoG | 1.00 | 34.238 | 0.782 |
| HiDDeN | 0.89 | 33.376 | 0.680 |

Table 12: Overall watermark removal performance on the DIV2K dataset. Detailed results of full threshold candidates are shown in Appendix A.3.2.

| Methods | PIMoG(th=0.6) | | | HiDDeN(th=0.6) | | | StegaStamp(th=0.57) | | | RivaGAN(th=0.625) | | | CIN(th=0.6) | | |
|---------|------|--------|--------|------|--------|--------|------|--------|--------|------|--------|--------|------|--------|--------|
| | SR↑ | SSIM↑ | PSNR↑ | SR↑ | SSIM↑ | PSNR↑ | SR↑ | SSIM↑ | PSNR↑ | SR↑ | SSIM↑ | PSNR↑ | SR↑ | SSIM↑ | PSNR↑ |
| WmAttacker | 0 | 0.739 | 26.853 | 0.34 | 0.64 | 23.954 | 0 | 0.71 | 26.2 | 0.15 | 0.536 | 19.978 | 0.01 | 0.644 | 23.031 |
| WmRobust | 0.21 | 0.809 | 33.04 | 0.2 | 0.814 | **34.525** | 0.1 | 0.845 | 33.858 | 0.37 | 0.826 | 31.803 | 0.33 | 0.810 | 34.847 |
| JPEG | 0 | **0.897** | **36.37** | 0.42 | 0.822 | 33.522 | 0 | 0.867 | **35.976** | 0.12 | 0.771 | 29.76 | 0.01 | **0.850** | **35.727** |
| Gaussian | 0 | 0.399 | 26.477 | 0.72 | 0.444 | 26.509 | 0 | 0.365 | 26.496 | 0.01 | 0.578 | 26.954 | 0 | 0.362 | 26.479 |
| GaussianBlur | 0 | 0.725 | 30.154 | 0.05 | 0.62 | 27.565 | 0 | 0.713 | 30.123 | 0.01 | 0.526 | 24.617 | 0 | 0.735 | 30.952 |
| **Ours** | **1** | 0.828 | 33.042 | **0.96** | **0.841** | 33.257 | **1** | **0.853** | 33.54 | **1** | 0.841 | **32.656** | **0.45** | 0.811 | 34.744 |

## A.3 ADDITIONAL EXPERIMENTAL RESULTS

### A.3.1 DETECTION EVASION ON DIV2K

### A.3.2 WATERMARK REMOVAL AGAINST RELATED WORKS

We provide watermark removal attack results as shown in 6 - 15 on COCO (Lin et al., 2014b) dataset and shown in 16- 19 on DIV2K (Agustsson & Timofte, 2017) dataset, including the attack success rates and visual metrics corresponding to all watermark detection thresholds. Additionally, we provide examples of attack results against other algorithms, as shown in 40, 38.

### A.3.3 WATERMARK FORGERY AGAINST RELATED WORKS

We provide detailed spoof attack results as shown in 20- 29, including the attack success rates and visual metrics corresponding to all watermark detection thresholds. Additionally, we provide examples of attack results against other algorithms, as shown in 39, 41.

### A.3.4 ABLATION STUDY FOR WATERMARK REMOVAL

We provide detailed ablation study for evasion attack results as shown in 30- 33, including the attack success rates and visual metrics corresponding to all watermark detection thresholds.

### A.3.5 ABLATION STUDY FOR WATERMARK FORGERY

We provide detailed ablation study for spoofing attack results as shown in 34- 37, including the attack success rates and visual metrics corresponding to all watermark detection thresholds.

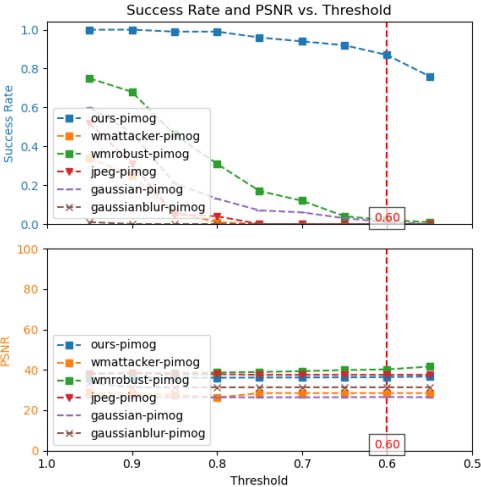

Figure 6: The detailed success rate of PIMoG evasion attacks and the corresponding PSNR visual metric.

Figure 7: The detailed success rate of PIMoG evasion attacks and the corresponding SSIM visual metric.

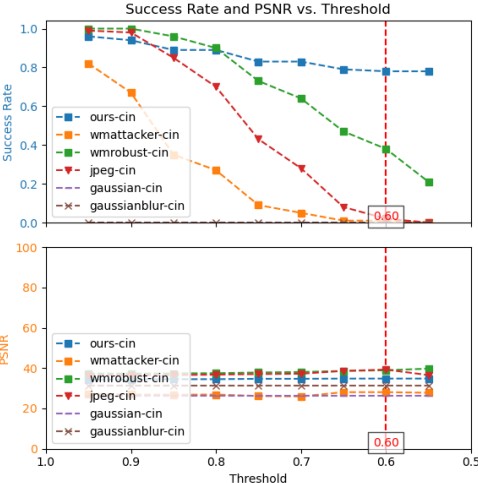

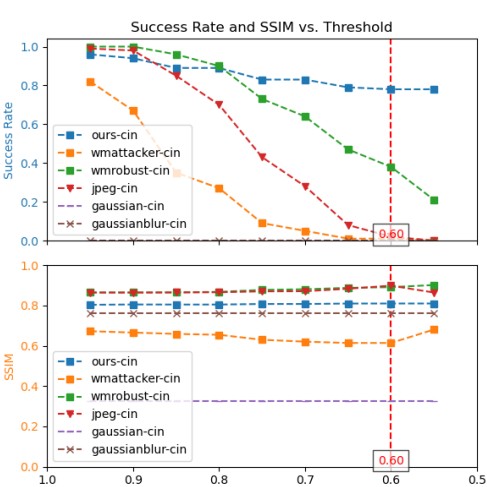

Figure 8: The detailed success rate of CIN evasion attacks and the corresponding PSNR visual metric.

Figure 9: The detailed success rate of CIN evasion attacks and the corresponding SSIM visual metric.

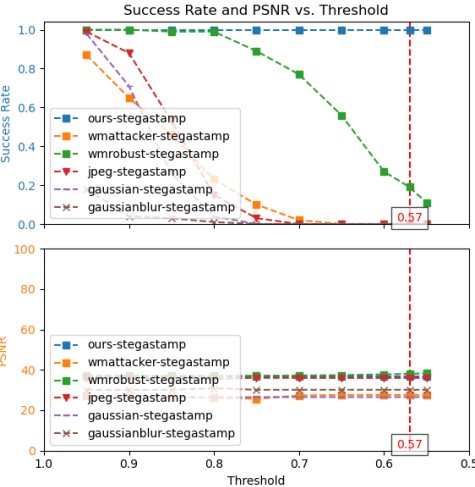

Figure 10: The detailed success rate of StegaStamp evasion attacks and the corresponding PSNR visual metric.

Figure 11: The detailed success rate of StegaStamp evasion attacks and the corresponding SSIM visual metric.

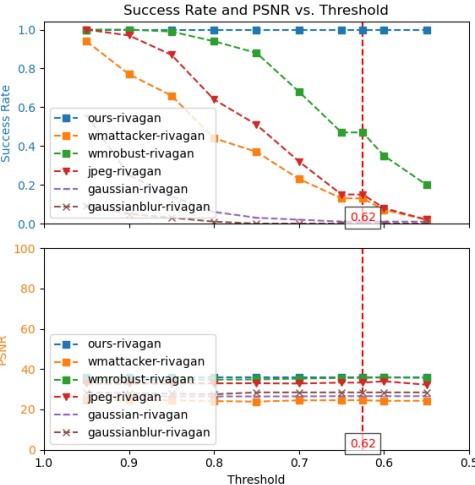

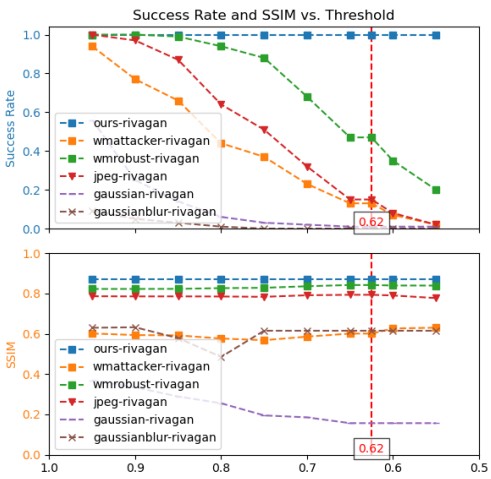

Figure 12: The detailed success rate of RivaGAN evasion attacks and the corresponding PSNR visual metric.

Figure 13: The detailed success rate of RivaGAN evasion attacks and the corresponding SSIM visual metric.

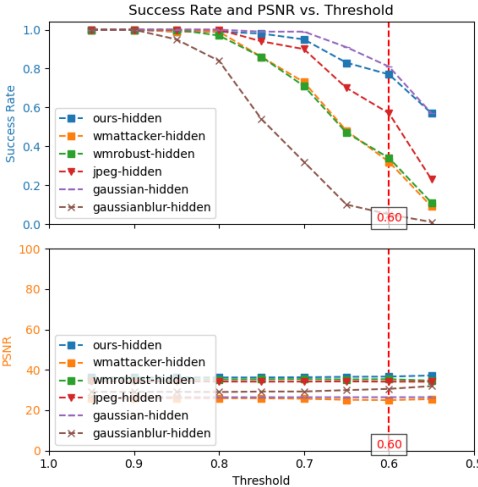

Figure 14: The detailed success rate of HiDDeN evasion attacks and the corresponding PSNR visual metric.

Figure 15: The detailed success rate of HiDDeN evasion attacks and the corresponding SSIM visual metric.

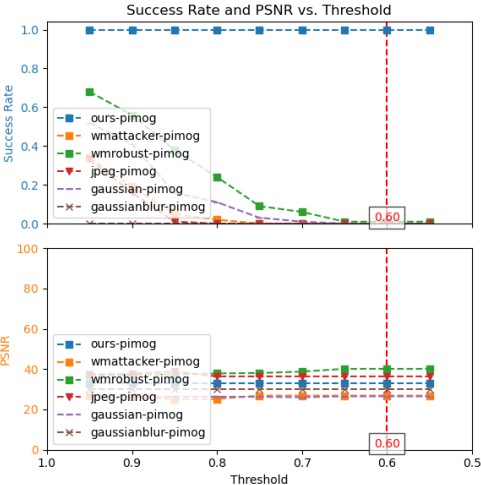

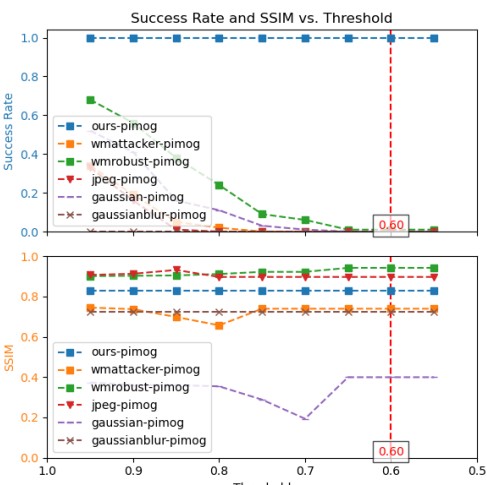

Figure 16: The detailed success rate of PIMoG evasion attacks and the corresponding PSNR visual metric on DIV2K dataset.

Figure 17: The detailed success rate of PIMoG evasion attacks and the corresponding SSIM visual metric on DIV2K dataset.

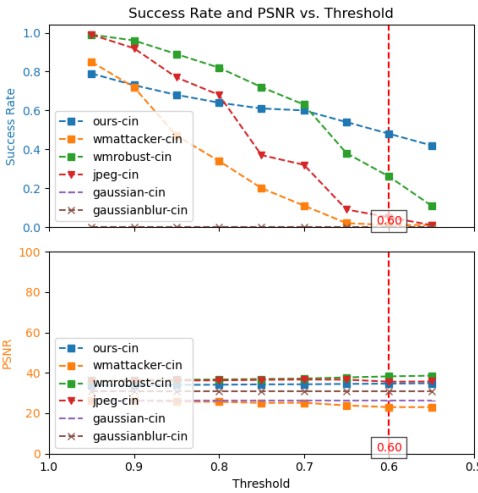

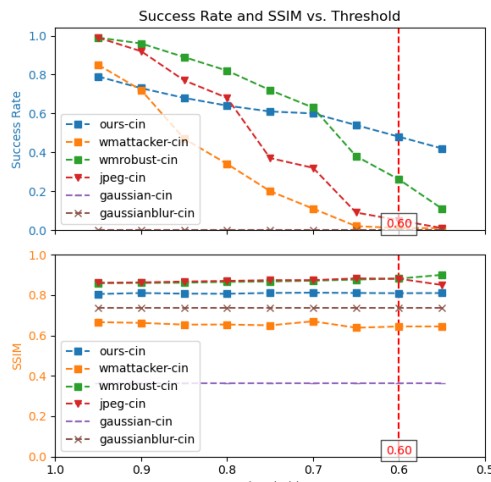

Figure 18: The detailed success rate of CIN evasion attacks and the corresponding PSNR visual metric on DIV2K dataset.

Figure 19: The detailed success rate of CIN evasion attacks and the corresponding SSIM visual metric on DIV2K dataset.

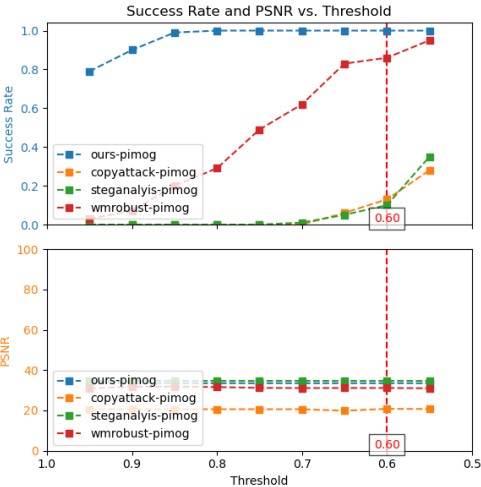

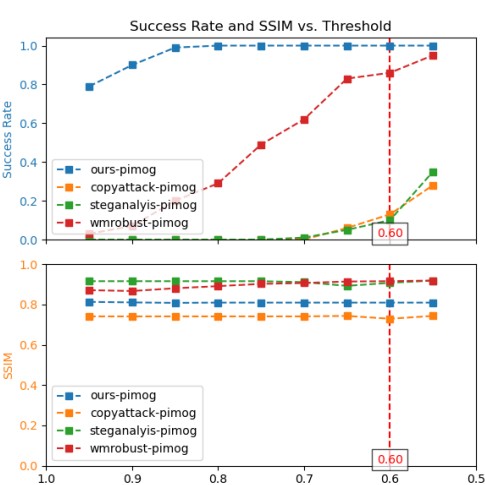

Figure 20: The detailed success rate of PIMoG forgery attacks and the corresponding PSNR visual metric.

Figure 21: The detailed success rate of PIMoG forgery attacks and the corresponding SSIM visual metric.

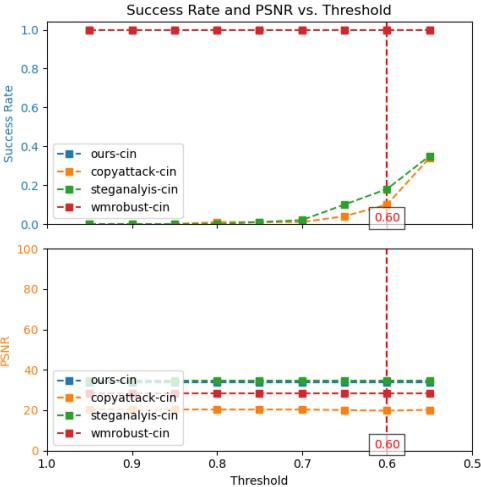

Figure 22: The detailed success rate of CIN forgery attacks and the corresponding PSNR visual metric.

Figure 23: The detailed success rate of CIN forgery attacks and the corresponding SSIM visual metric.

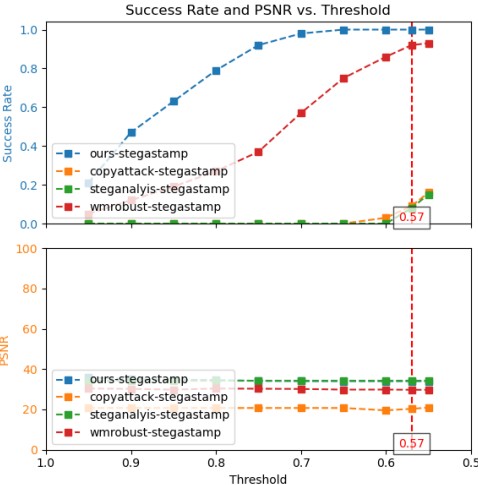

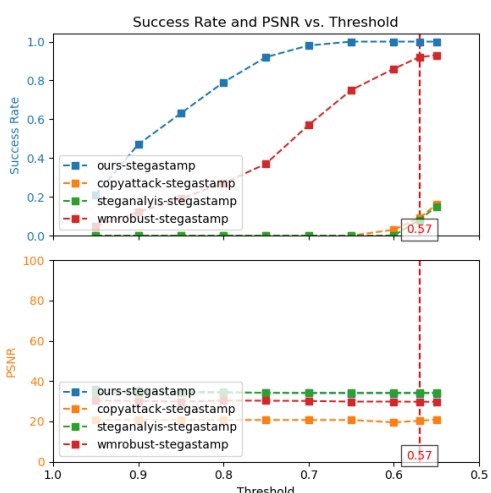

Figure 24: The detailed success rate of StegaStamp forgery attacks and the corresponding PSNR visual metric.

Figure 25: The detailed success rate of StegaStamp forgery attacks and the corresponding SSIM visual metric.

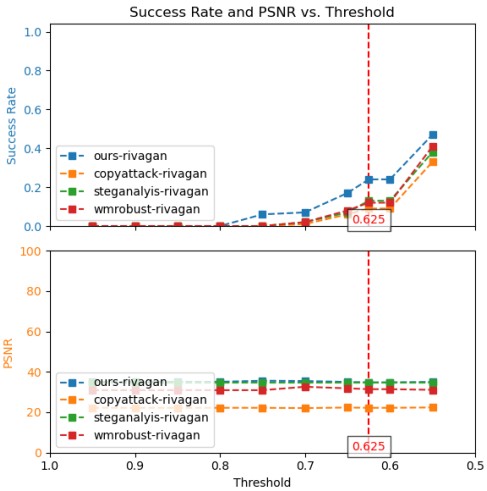

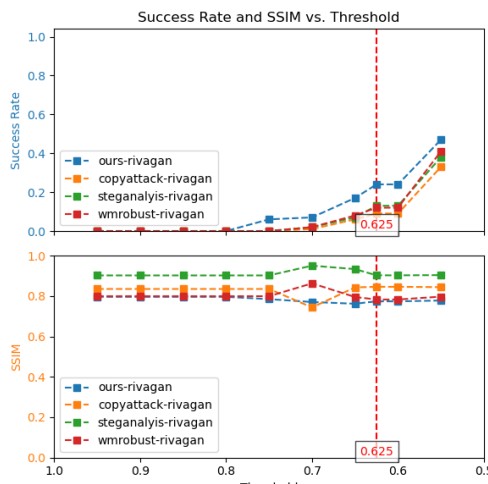

Figure 26: The detailed success rate of RivaGAN forgery attacks and the corresponding PSNR visual metric.

Figure 27: The detailed success rate of RivaGAN forgery attacks and the corresponding SSIM visual metric.

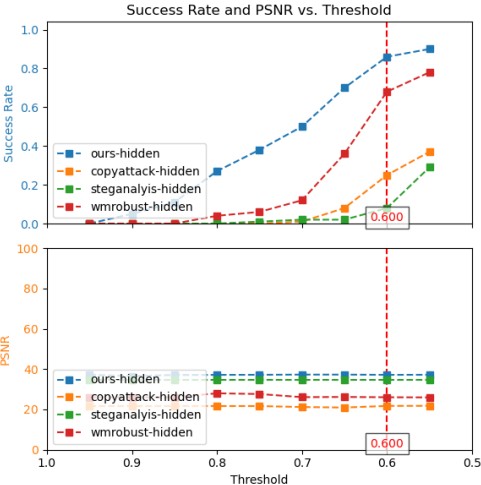

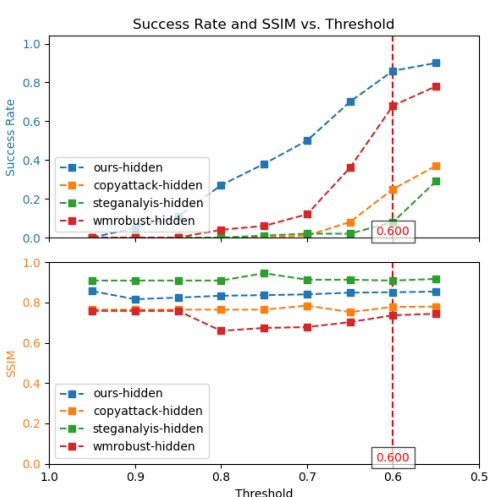

Figure 28: The detailed success rate of HiDDeN forgery attacks and the corresponding PSNR visual metric.

Figure 29: The detailed success rate of HiDDeN forgery attacks and the corresponding SSIM visual metric.

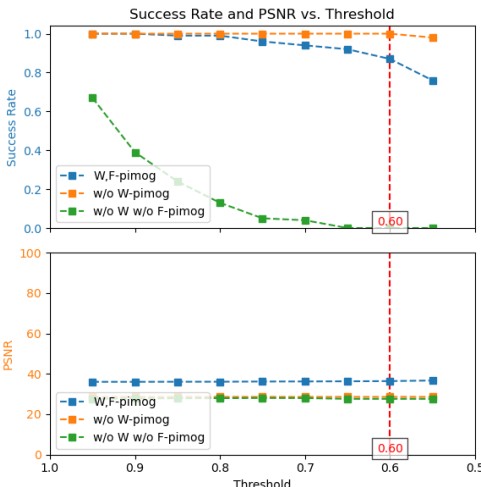

Figure 30: The detailed success rate of PIMoG evasion attacks and the corresponding PSNR visual metric.

Figure 31: The detailed success rate of PIMoG evasion attacks and the corresponding SSIM visual metric.

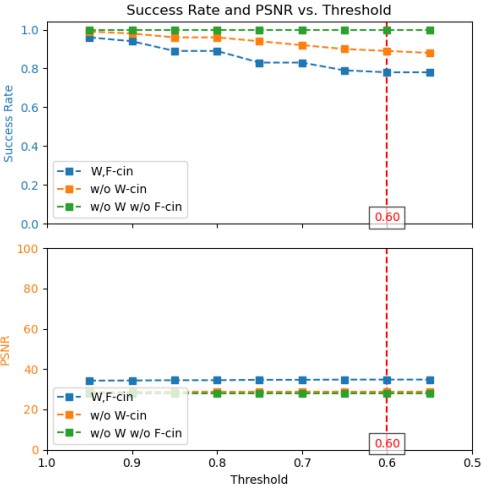

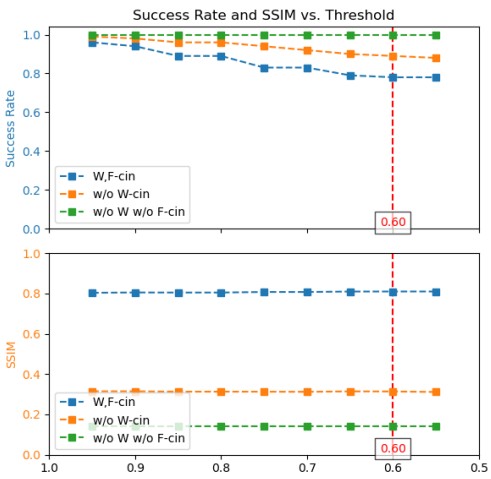

Figure 32: The detailed success rate of CIN evasion attacks and the corresponding PSNR visual metric.

Figure 33: The detailed success rate of CIN evasion attacks and the corresponding SSIM visual metric.

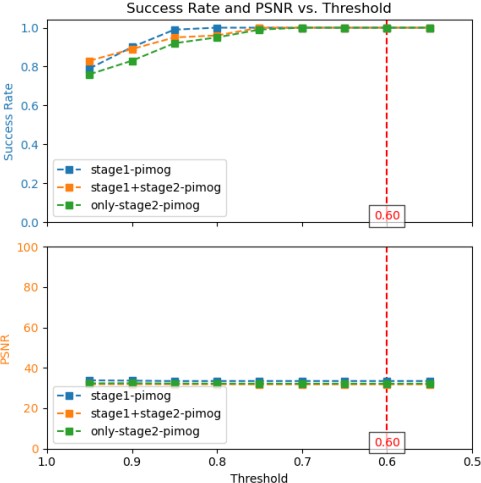

Figure 34: The detailed success rate of PIMoG forgery attacks and the corresponding PSNR visual metric.

Figure 35: The detailed success rate of PIMoG forgery attacks and the corresponding SSIM visual metric.

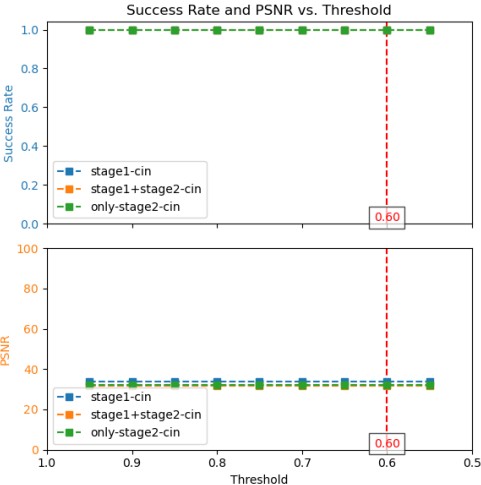

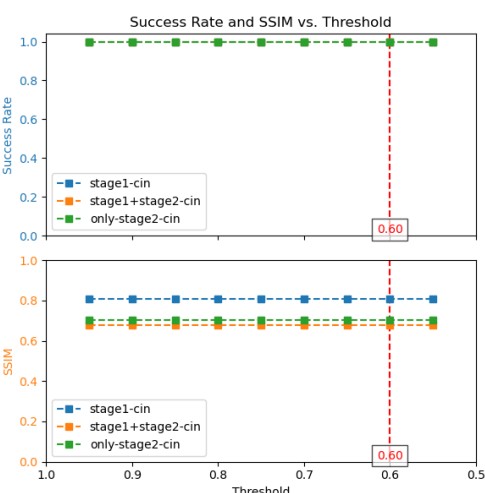

Figure 36: The detailed success rate of CIN forgery attacks and the corresponding PSNR visual metric.

Figure 37: The detailed success rate of CIN forgery attacks and the corresponding SSIM visual metric.

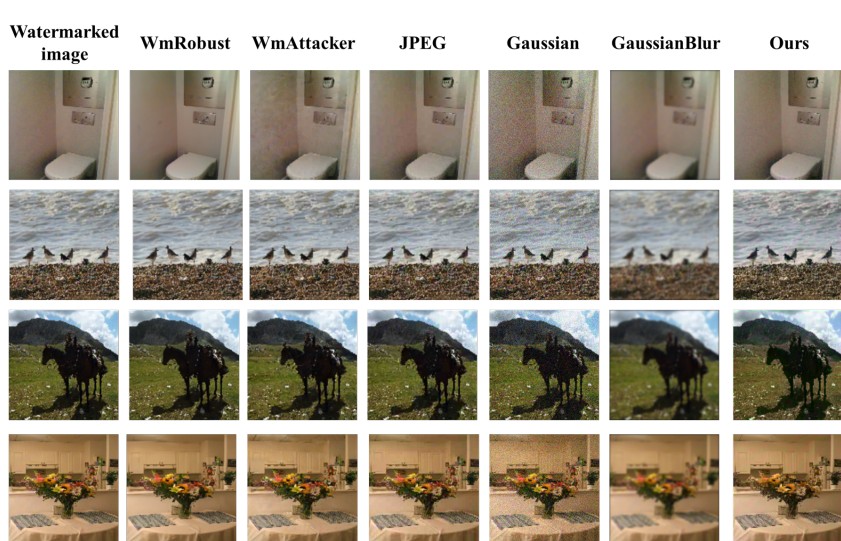

Figure 38: Examples of watermark removal via our removal attack on PIMoG

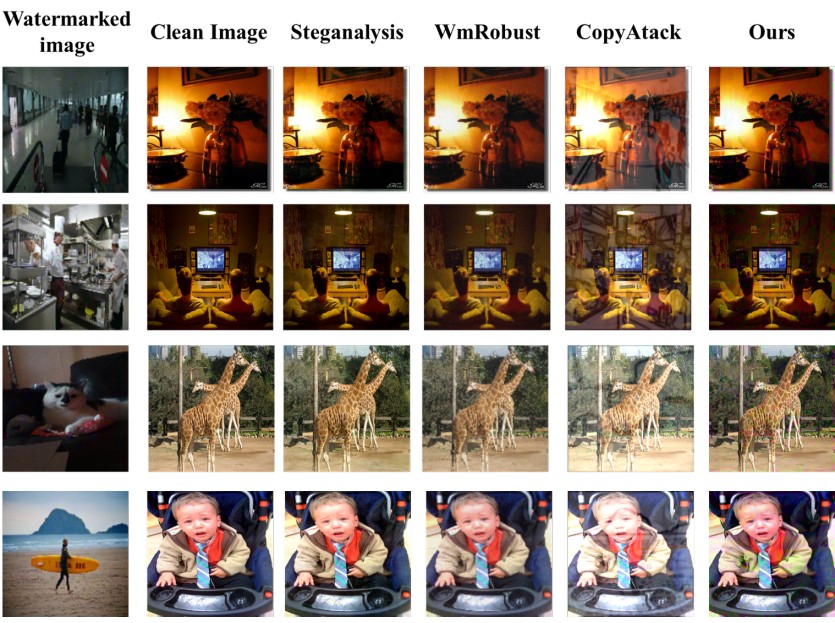

Figure 39: Examples of watermark spoofing via our forgery attack on PIMoG.

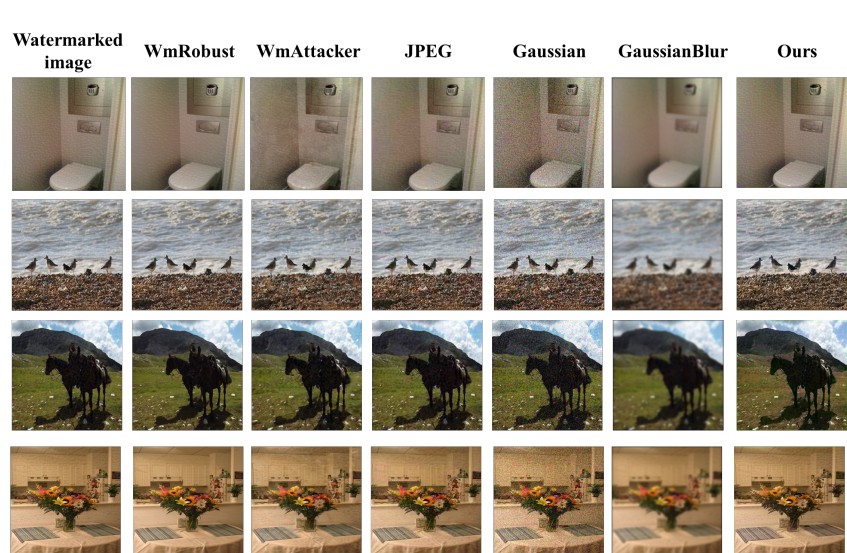

Figure 40: Examples of the evasion attack against HiDDeN.

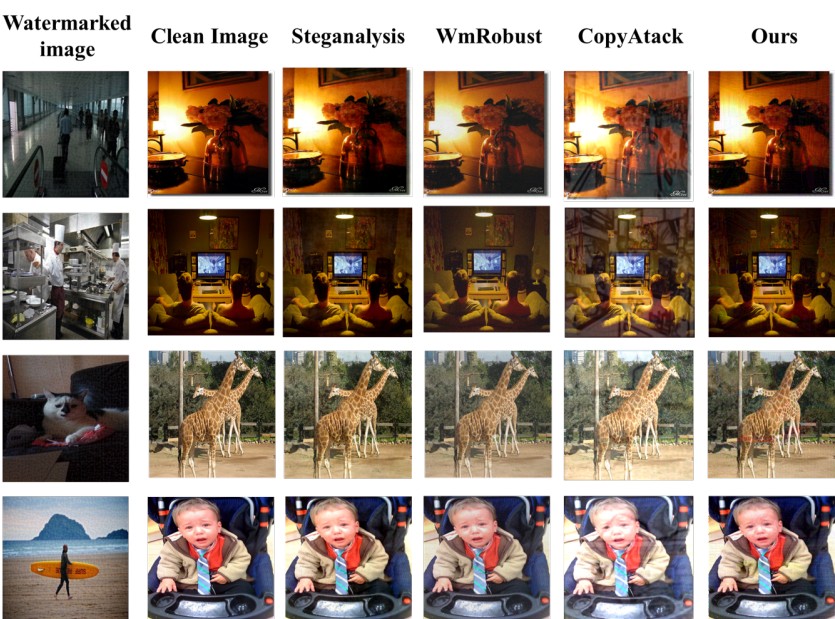

Figure 41: Examples of the spoof attack against HiDDeN.

### A.3.6 ALIGNMENT BETWEEN THE WATERMARK INFORMATION AND THE SEMANTIC CONTENT

To demonstrate that certain watermarking models consider the alignment between semantics and watermark information during the decoding process, we conducted an experiment on forged watermarks.

$I_c$ represents clean images, and $I_s$ represents forged images. Embedding the watermark into $I_o$ using RivaGAN to obtain $I_{wm}$ then obtaining the watermark $W_{GT}$ by $I_o - I_{wm}$. Varying parameters $a = \{1 \rightarrow 1.3\}$, and $b = \{0 \rightarrow 0.3\}$ to get $I_s$ and compute the average bit accuracy.

Table 13: Conducting forgery watermark overlay experiments on RivaGAN. $I_o$ denotes the original image, $I_c$ denotes clean images, and $I_s$ denotes forged images. A watermark is embedded into $I_o$ using RivaGAN to obtain $I_{wm}$, and the ground-truth watermark is computed as $W_{GT} = I_o - I_{wm}$. The results demonstrate that it requires a matched watermark and its corresponding host image to correctly extract the watermark, highlighting the difficulty of launching forged attacks against RivaGAN.

| Strategy | Avg. Bit ACC↑ |
| --- | --- |
| $I_s = I_c + W_{GT} \cdot 1 + I_o \cdot 0$ | 0.50 |
| $I_s = I_c + W_{GT} \cdot 1.1 + I_o \cdot 0$ | 0.51 |
| $I_s = I_c + W_{GT} \cdot 1.1 + I_o \cdot 0.1$ | 0.57 |
| $I_s = I_c + W_{GT} \cdot 1.2 + I_o \cdot 0.1$ | 0.59 |
| $I_s = I_c + W_{GT} \cdot 1.2 + I_o \cdot 0.2$ | 0.61 |
| $I_s = I_c + W_{GT} \cdot 1.3 + I_o \cdot 0.2$ | 0.60 |
| $I_s = I_c + W_{GT} \cdot 1.3 + I_o \cdot 0.3$ | 0.71 |

### A.4 ALGORITHM

Our detailed algorithm as shown:

---

**Algorithm 1** Evasion Attack

---

**Input:** watermarked image $I_{wm}$, step $m$, feature extracter $\mathcal{F}$, clustering algorithm $\mathcal{C}$, optimizer $\mathcal{O}$, objective function $\mathcal{L}$, perturbation budget $\epsilon_1$
**Output:** attacked watermarked image $I_a$
// find the k clusters with the fewest samples within the cluster, and transform the channel feature indices within the clusters into weights for locating information leakage.
$\mathcal{W} \xleftarrow{\mathcal{C}} \mathcal{F}(I_{wm})$
$\delta \leftarrow noise$
**for** $i = 1$ **to** $m - 1$ **do**
    $\delta \leftarrow \mathcal{O}(\delta, -\nabla_\delta \mathcal{L}(\mathcal{W} \cdot \mathcal{F}(I_{wm}), \mathcal{W} \cdot \mathcal{F}(I_{wm} + \delta))))$
    **if** $||\delta||_\infty > \epsilon_1$ **then**
        $\delta \leftarrow \delta \cdot \frac{\epsilon_1}{||\delta||_\infty}$
    **end if**
**end for**
**return** $I_a = I_{wm} + \delta$

---

**Algorithm 2** Spoof Attack

---

**Input:** watermarked image $I_{wm}$, clean image $I'$, step $m$, feature extracter $\mathcal{F}$, clustering algorithm $\mathcal{C}$, optimizer $\mathcal{O}$, objective function $\mathcal{L}$, perturbation budget $\epsilon_1, \epsilon_2$
**Output:** attacked watermarked image $I_a$
// find the k clusters with the fewest samples within the cluster, and transform the channel feature indices within the clusters into weights for locating information leakage.
$\mathcal{W} \xleftarrow{\mathcal{C}} \mathcal{F}(I_{wm})$
$\delta \leftarrow noise$
// Stage-I
**for** $i = 1$ **to** $m - 1$ **do**
$\quad \delta \leftarrow \mathcal{O}(\delta, \nabla_\delta - \mathcal{L}(\mathcal{W} \cdot \mathcal{F}(I_{wm}), \mathcal{W} \cdot \mathcal{F}(I_{wm} + \delta))))$
$\quad$ **if** $||\delta||_\infty > \epsilon_1$ **then**
$\qquad \delta \leftarrow \delta \cdot \frac{\epsilon_1}{||\delta||_\infty}$
$\quad$ **end if**
**end for**
// Stage-II
$\delta_s \leftarrow \delta$
**for** $i = 1$ **to** $m - 1$ **do**
$\quad \delta_s \leftarrow \mathcal{O}(\delta_s, \nabla_{\delta_s} \mathcal{L}((1 - \mathcal{W}) \cdot \mathcal{F}(I_{wm} + \delta), (1 - \mathcal{W}) \cdot \mathcal{F}(I' + \delta_s)))$
$\quad \hat{\delta} \leftarrow \delta_s$
$\quad$ **if** $||\hat{\delta}||_\infty > \epsilon_2$ **then**
$\qquad \hat{\delta}_s \leftarrow \hat{\delta} \cdot \frac{\epsilon_2}{||\hat{\delta}||_\infty}$
$\quad$ **end if**
**end for**
**return** $I_a \in \{I' - \delta, I' - \delta + \delta_s\}$

---

### A.5 THE USE OF LARGE LANGUAGE MODELS

Large language models were used solely for writing revision and language polishing; all research methods, experimental designs, and result analyses were independently conducted by the authors.

