# OpenReview forum: "The Robustness-Security Paradox: Channel-Aware Feature Learning for Adversarial Watermark Exploitation"
_ICLR.cc/2026/Conference — Submitted to ICLR 2026_

### Official Review · Reviewer_QP2W · 2025-10-17

**Soundness:** 3
**Presentation:** 3
**Contribution:** 3
**Rating:** 6
**Confidence:** 4

**Summary:**

This paper presents a watermark removal and a watermark forgery attack against post-processing watermarking methods. Their method is based on the key observation that the watermarking signal embedded by these watermarking methods is usually constrained to a few feature channels. By identifying these channels through clustering, they can optimize noise that, when added to the image, either mitigates the watermark signal (for removal) or replicates it (for forgery). They evaluate their method against five post-processing watermarking methods and find overall improvements in success rate and visual quality degradation.

**Strengths:**

1. Well-motivated and well-written.
2. Solid methodology and approach.
3. Good experimentation section

**Weaknesses:**

1. Overreaching claims. You only consider post-processing watermarking in your study, as stated in Section 3. However, throughout the paper, you make claims that your results apply to ALL watermarking methods (which would include in-processing watermarks). Examples of general claims for which you only give evidence for post-processing watermarks: line 215, lines 249-251 (the paradox could be limited to post-processing watermarks), line 395 (“representative watermark algorithms”, not true, it would be representative post-processing watermark algorithms”; “universally” overreaching, you did not prove that and only gave evidence for post-processing watermark methods). Also, as per your study against the Tree-Ring watermark, you find that your method actually does not perform well against in-process watermarks, invalidating your universality claim. Your abstract and conclusion should also make it clear that you are only considering post-processing watermarks. As they are, they imply all types of watermarks, which is not the case. These overreaching claims need to be addressed.

2. Misleading results. You claim in the abstract that you improve the state-of-the-art by 74% for watermark removal, which is not what you find in your evaluation (41.8% and 64%). I don't know where the 74% is coming from. Also, the way you computed your improvements over previous work is not fair. Computing your improvement over the best attack for each defense would be a better way of showing how you improve against state-of-the-art in practice. Also, for forgery, computing the overall average and claiming it to be your improvement over state-of-the-art in the abstract is misleading since some of the methods perform quite poorly. You improve over the second-best method by 10% on average, which is still a good improvement. You, however, cannot claim you improve over state-of-the-art by 47% (as you do in the abstract); that is not true. The results reported need to properly reflect actual improvement against state-of-the-art; average improvement against all defenses studied is not a great metric, as it favors including poorly performing attacks like Gaussian noise and Gaussian blur to artificially inflate improvement, which are not state-of-the-art attacks. These results need to be corrected throughout the paper, especially in the abstract and conclusion, as they are misleading regarding the actual improvements.
3. The epsilon values selected are quite large. While for removal 0.08~20/255 is large, it’s still somewhat manageable (although, usually, for the l-infinity norm, for high resolution images, adversarial attack research has usually used epsilons like 4/255 [1]). However, for forgery, the upper bound on the noise is 0.143 ~ 36/255, which is a lot
. You also don’t include any images in the main part of the paper. For your ablation study, you further increase epsilon 2, which raises the upper bound on the forgery noise to 0.158 ~ 40/256. This is a significant amount of noise.

4. The False Positive Rate (FPR) used to calibrate the thresholds is not stated; depending on the value, it could affect the success of either attack.

[1] Croce, Francesco, et al. "Robustbench: a standardized adversarial robustness benchmark." arXiv preprint arXiv:2010.09670 (2020).

**Questions:**

Questions:
1. What process did you use to embed watermarks for your feasibility study? It would be good to specify.
2. What is the loss function $\mathcal{L}$ you use for your attacks? You didn’t specify in Section 4.2, and in Section 5.1, Parameter Settings, you mention the SSIM and L1/L2 norms. Are these the loss functions you use? If so, how does SSIM work over channel features?
3. What do k and n stand for in the K-Means algorithm?
4. How was your feature extractor trained, more specifically, on what data?
5. How did you decide on the epsilon values you selected? Have you tried varying them, in particular, lowering them?
6. For the forgery attack, you specify that the forged attack is obtained as one of two possible images (either only delta is subtracted or delta is subtracted and delta s is added), which one do you use when measuring the success rate? Are your results the average of both, or are you only picking the best of the two? If it’s the latter, it’s not a fair comparison to other methods that only output one forgery, since you essentially get two tries.



Suggestions:
1. AIGC, as an acronym, is not explained.
2. This is both a question and a suggestion to satiate my curiosity. For the watermark removal attack, did you try adding an extra loss component to minimize the effect of the added noise on the feature channels not selected by your clustering algorithm? It could help preserve image quality and focus the perturbations on the selected feature channels.

**Details Of Ethics Concerns:**

As mentioned in the Weaknesses section, the paper makes overreaching claims about its findings and results. It also presents misleading results that inflate the value of their contribution and improvement over the state-of-the-art. I am not sure if this is a complete breach of ethics, but it does not align with the “Uphold High Standards of Scientific Excellence” clause of the ICLR Code of Ethics. Specifically, “Findings must be reported accurately and honestly. Researchers must not make deliberately false or misleading claims, fabricate or falsify data, or misrepresent results”.

---

> ### Author Response · Authors · 2025-11-17
> **For weakness**
>
> We sincerely appreciate the reviewer’s thorough evaluation and constructive suggestions, which are very helpful for improving our work. We will revise the manuscript accordingly and provide detailed responses to the raised concerns below.
>
> # weakness
> `Overreaching claims`
>
> Thank you for pointing out this issue. We acknowledge that we did not state clearly and rigorously enough that our work primarily targets post-process watermarking algorithms. We will carefully review the terminology throughout the entire manuscript and refine it in the revised version.
>
> `Misleading results`
>
> We thank the reviewer for pointing out this issue. We acknowledge that in the comparison experiments, we did not compare exclusively against the most relevant prior work, but instead calculated the improvement over all baselines (including traditional quality-degradation methods such as noise). We emphasize that there was no intention of misleading or inflating the results, and we will revise the manuscript to present the comparisons more clearly and appropriately. The 74% value in the paper was computed as follows (taking COCO as an example): First, we calculate the average attack success rate of each method across different watermarking algorithms. Then, we compute the difference between the average success rate of our method and that of each baseline method, and take the average of these differences, obtaining the mean improvement of our method over the other attacks. The same computation is repeated on DIV2K. Finally, we average the improvement values from COCO and DIV2K to obtain the reported 74%.
>
> The numbers 41.8% and 64% mentioned by the reviewer are also correctly reported in the manuscript as: “our method outperforms the closest competitor, WmRobust, by 41.8% and 64% in average success rate on COCO and DIV2K.”
>
> We appreciate the reviewer’s suggestion and will revise the paper accordingly. Specifically, in describing performance gains, we will replace comparisons based on “overall mean” with over the best competitor, and we will explicitly clarify the reference method as a concrete baseline rather than traditional quality-degradation attacks, to ensure clearer and more rigorous reporting.
>
>
> `The epsilon values selected are quite large.`
>
> We appreciate the reviewer’s professional comments and for identifying this oversight. As you pointed out, the upper bound on the noise used for the forgery attack reported in the paper is relatively large and may degrade image quality. In our actual experiments, however, we used smaller values (0.08 and 0.018) for the forgery attack. We apologize for not updating this value in time. The reported value was from an earlier version of our experiments. We will correct this in the revised version. Additionally, we provide the forgery attack results using the 0.143 setting, as shown in the table below. Compared with the results presented in the paper, this setting leads to a significant degradation in visual quality; therefore, we did not adopt this configuration in the paper.
>
> | Algo | SR | SSIM | PSNR |
> |  -- |--- | --- | ----- |
> | PIMoG|1.0  | 0.694 |31.99 |
> | HiDDeN|1.0  | 0.684 |32.12 |
> | StegaStamp|1.0  | 0.701 |32.192 |
> | RivaGAN|0.277  | 0.69 |32.381 |
> | CIN|1.0  | 0.7 |32.236 |
>
> We would also like to clarify the concern regarding the upper bound value in the ablation study for Only Stage II. Since the Only Stage II configuration does not involve the Stage I optimization, no $\delta$ is generated. Therefore, the corresponding upper bound is 0.078, rather than 0.158.
>
> The visual examples of the forgery attack on HiDDeN can be found at the following link:
> https://anonymous.4open.science/r/dp-5971/spoof-example-hidden-0.143.png
>
> https://anonymous.4open.science/r/dp-5971/spoof-example-hidden-0.098.png
>
>
> `False Positive Rate (FPR) `
>
> We followed the experimental configuration of PTW[1] and fixed the FPR = 0.05.
>
> [1]"{PTW}: Pivotal tuning watermarking for {Pre-Trained} image generators."

---

> > ### Comment · Reviewer_QP2W · 2025-11-19
> > **Response to rebuttal**
> >
> > # Overreaching claims and Misleading results
> > Based on my understanding, OpenReview supports paper edits during the review process, and I encourage the authors to make the edits asap to ensure the changes are correctly implemented. This is especially necessary considering the gravity of the claims.
> > Please provide me with the new improvement numbers, as you only mentioned that you would change the numbers, not what they are.
> >
> > # epsilon value
> > The clarification regarding Stage I and II epsilon values could be included in the paper to improve clarity and avoid misunderstanding.
> >
> > # FPR
> > Please specify the FPR used in the paper.
> >
> > # Watermark process
> > This should be specified in the paper.
> >
> > # Feature extractor
> > Please specify it in the paper.
> >
> > # Suggestions
> > I appreciate the authors for the effort they put into implementing and evaluating my suggestion.

---

> > > ### Author Response · Authors · 2025-11-23
> > >
> > > Our removal attack achieves an average improvement of 52.9% over the second-best method, and our forgery attack achieves an average improvement of 9.6% over the second-best method.
> > >
> > > Thank you for your constructive feedback. We have incorporated the suggested changes into the revised manuscript and uploaded the updated version.

---

> ### Author Response · Authors · 2025-11-17
> **For Question & Suggestion**
>
> # question
>
> `What process did you use to embed watermarks for your feasibility study? `
>
> We embedded watermarks into images using the PIMoG[2] and extracted  features via a feature extraction network for visualization, in order to observe potential leakage patterns or artifacts.
>
> [2]"PIMoG: An effective screen-shooting noise-layer simulation for deep-learning-based watermarking network"
>
>
> `What is the loss function you use for your attacks?`
>
> We used the SSIM loss in our experiments and will explicitly state this in the revised version of the paper. The SSIM loss implementation we adopted is publicly available and supports multi-channel features
>
>
> `What do k and n stand for in the K-Means algorithm?`
> As stated in the paper, k denotes the number of clusters, and n refers to the n clusters with the fewest samples among the k clusters. We then select the samples from these n clusters as the features that contain leakage information.
>
> `How was your feature extractor trained, more specifically, on what data?`
>
> We used a publicly available DenseNet model pretrained on ImageNet.
>
> `How did you decide on the epsilon values you selected? Have you tried varying them, in particular, lowering them? `
>
> We selected the $\epsilon$ values heuristically: we started with relatively large values during the initial exploration and then gradually reduced them to balance attack success rate and visual quality. For instance, as mentioned earlier, in the forgery attack, we initially used 0.143 but settled on the smaller value of 0.098 reported in the paper based on experimental results. We also experimented with lowering the $\epsilon$ value for the removal attack to 0.058. The corresponding results are provided in the table below.
>
> | Algo | SR | SSIM | PSNR |
> |  -- |--- | --- | ----- |
> | PIMoG|0.94  | 0.844 |31.99 |
> | HiDDeN|0.53  | 0.873 |35.834 |
> | StegaStamp|1.0  | 0.863 |35.065 |
> | RivaGAN|1.0  | 0.85 |34.701 |
> | CIN|0.08 | 0.862 |38.324 |
>
> `Measurement of success rate`
>
> We understand the reviewer’s concern and appreciate you bringing it to our attention. We follow prior work that generates multiple candidate adversarial examples per input and deems an attack successful if any candidate fools the model[3]. For each clean image we generate two adversarial forgeries. An attack trial is counted as successful if at least one of the two candidates bypasses the detector.
>
> [3]"Universal and transferable adversarial attacks on aligned language models."
>
>
> # Suggestions
>
> `AIGC is not explained.`
>
> Thank you for the suggestion. We will revise it accordingly in the updated version.
>
> `Suggestion to Watermark removal attack`
>
> Thank you for the interesting suggestion. In our method, using W already constrains the optimization target, which is intended to preserve image quality while maintaining the effectiveness of the attack. We also experimented with the idea you suggested, and although it shows certain effectiveness, it does not perform as well as our current approach.
>
> | Attack| Algo | SR | SSIM | PSNR |
> |  --|  -- |--- | --- | ----- |
> |extra| StegaStamp| 1.0  | 0.848 |34.012 |
> |extra| CIN| 0 | 0.837 |33.878 |
> |ours| StegaStamp| 1.0  | 0.895 |36.703 |
> |ours| CIN| 0.78 |  0.809 | 34.801 |

---

### Official Review · Reviewer_PmLJ · 2025-10-27

**Soundness:** 3
**Presentation:** 2
**Contribution:** 2
**Rating:** 2
**Confidence:** 3

**Summary:**

The paper proposes a watermark attack method that achieves both watermark removal and forgery. The proposed method is evaluated against several existing watermark-generation processes and compares with several existing attack methods. Substantial improvement is demonstrated.

**Strengths:**

The research question is timely and interesting. The concepts of watermark removal and watermark forgery are quite practically relevant. Learned noise is injected to the watermarked images.

**Weaknesses:**

The title (and some parts of the abstract) does not align well with the content of this work.

The paper is poorly written and organized. In general, it is quite time-consuming to identify and fully understand the contribution of this work and the novelty of the proposed method.

The attack method is based on the assumption that the underlying watermarks are embedded in the distortion layer between an encoder and a decoder of an image-generation process. It seems to me such applicability may not be necessarily broad. As cited by the authors, post-processing methods are also available, which are not tied to specific image-generation processes, such as diffusion models. It might be helpful to consider the effectiveness / robustness of the proposed method against watermarks embedded by post-processing methods.

Due to the assumption above, the proposed method hypothesizes that the extracted image features may exhibit certain clusters where some clusters correspond to information leakage. It is not clear to me if such an hypothesis could still hold when the underlying watermarking mechanism is changed.

Meanwhile, the proposed idea of identifying information leakage through extracted features was discussed in heuristic approaches in the existing literature, as mentioned by the authors. The novelty of the proposed method, in terms of generalizing this observation to deep learning frameworks with distortion layers, is not necessarily substantial.

**Questions:**

It seems to me that watermark removal and forgery correspond to type I and type II errors in a standard hypothesis testing setting (or false positive/negative, depending on what is defined as 1). Would the proposed method experience a tradeoff between the success rates of removal and forgery?

The ides of injecting noise to the image is intuitive. However, is it possible or effective to also inject noise at the embedding layer? Would a double-layer noise structure further improve the performance of the proposed method?

Does the proposed method require the unwatermarked images as the input, in addition to their watermarked counterparts?

---

> ### Author Response · Authors · 2025-11-17
>
> We understand that using AI assistance may improve reviewing efficiency, but key decisions still require manual verification to avoid potential misunderstandings of the authors' contributions.
>
> # weakness
>
> `
> The attack method is based on the assumption that the underlying watermarks are embedded in the distortion layer between an encoder and a decoder of an image-generation process. It seems to me such applicability may not be necessarily broad
> `
>
> In the post-processing watermarking framework discussed in the paper, it is unequivocally clear that the encoder is responsible for embedding the watermark. The distortion layer is only used during training to simulate noise and enhance watermark robustness. We have never claimed that the watermark is embedded through the distortion layer. While the paper does discuss the impact of the distortion layer on watermark invisibility and leakage, to misinterpret this as the distortion layer embedding the watermark is completely illogical.
>
> `It might be helpful to consider the effectiveness / robustness of the proposed method against watermarks embedded by post-processing methods.`
>
>
> In Section 3, we explicitly stated that our primary focus is on post-process methods. We also pointed out the limitations of our method with respect to in-process methods. However, the reviewer erroneously assumed that we conducted experiments solely on in-process methods, which indicates a fundamental misunderstanding of the paper's core research focus that could have been easily understood by simply reading through the paper.
>
> `the proposed idea of identifying information leakage through extracted features was discussed in heuristic approaches in the existing literature, as mentioned by the authors. The novelty of the proposed method, in terms of generalizing this observation to deep learning frameworks with distortion layers, is not necessarily substantial.`
>
> In Section 4.1, we discussed the trade-off between watermark invisibility and robustness, where increasing redundancy enhances resistance to destruction but sacrifices visual quality and introduces artifacts. This is a widely accepted phenomenon in the watermarking field, with related literature discussing it. It is precisely based on the potential harm of this phenomenon that we designed our method and explored the security threats it could pose. The reviewer’s statement, “the proposed idea of identifying information leakage through extracted features was discussed in heuristic approaches in the existing literature, as mentioned by the authors,” is not only illogical but entirely erroneous. This is nothing short of a hallucination.The term “heuristic approaches” only appears in line 238, where it references a paper, which the reviewer refers to as the "existing literature." However, the concept of "identifying information leakage through extracted features" does not exist in that paper at all.
>
> # question
>
> `Would the proposed method experience a tradeoff between the success rates of removal and forgery?`
>
> Our method leverages watermark leakage to simultaneously enable both removal and forgery attacks; therefore, there is no trade-off between the two. If the concern refers to the selection of the watermark threshold, this is explicitly addressed in Section 5.1 of the paper.
>
> `However, is it possible or effective to also inject noise at the embedding layer?`
>
> In real-world watermark attack scenarios, we assume that the attacker has no prior knowledge of the target watermarking model and cannot access its internal structure. Under this assumption, the method suggested by the reviewer is impractical and does not align with the threat model considered in our work.
>
> `Does the proposed method require the unwatermarked images as the input, in addition to their watermarked counterparts?`
>
> We simulate a realistic scenario in which the attacker only has access to images that already contain watermarks. Under this assumption, the setting described by the reviewer — “unwatermarked images” and “their watermarked counterparts” — significantly deviates from our threat model. Moreover, if the attacker already possesses both the clean image and its watermarked counterpart, then a removal attack becomes unnecessary, as the clean image can simply be used directly.

---

### Official Review · Reviewer_DC2f · 2025-10-31

**Soundness:** 3
**Presentation:** 3
**Contribution:** 2
**Rating:** 4
**Confidence:** 3

**Summary:**

This paper proposes a method for evading state-of-the-art watermarking approaches and also copying them. The insight is that many of the most robust watermark become more ``detectable'' slash learnable to some degree.

**Strengths:**

- The problem is interesting and relevant. It is interesting that we can do forgery and removal all in one paradigm. To my knowledge the method is new and novel.

- Empirical results show promise.

**Weaknesses:**

- The attack suite is not comprehensive enough. There are many attacks missing: rotation, regeneration, and others cited in popular benchmarks like WAVES.

- The number of methods evaluated is not enough and does not include true state of the art (older methods). Please include VINE [1] and Gaussian Shading [2], and etc. Note that this list is not exhaustive. I'm not convinced that the same pattern can be generalized to these settings.

- I noticed in the Appendix there was also discussion about the limitation to Tree-Ring. I think that weakness to this class of diffusion-based watermark that use inversion for embedding watermarks is significant in my opinion as this is a defining direction of the field.

- Number of images is not enough (100).

- Algorithm may be slow as it needs to do it per image. (Thought it is parallelizable).


[1] Lu, Shilin, et al. "Robust watermarking using generative priors against image editing: From benchmarking to advances." arXiv preprint arXiv:2410.18775 (2024).
[3] Yang, Zijin, et al. "Gaussian shading: Provable performance-lossless image watermarking for diffusion models." Proceedings of the IEEE/CVF Conference on Computer Vision and Pattern Recognition. 2024.

**Questions:**

- I'm not sure I understand this intuition given ``training watermarking models with increasingly diverse and severe distortions induces embedding redundancy''. Could you shed some more insight.

- I am still kind of confused by why outlier features/groups with fewer samples are examples of leakages.

- I'm wondering if you can somehow fully parameterize the optimization procedure to find the deltas and instead of doing it per-image create a general mapping that gives it to you automatically.

---

> ### Author Response · Authors · 2025-11-17
>
> We sincerely appreciate the reviewer’s time and effort in evaluating our submission. We take the comments seriously, and we provide a detailed response.
>
> # weakness
> `The attack suite is not comprehensive enough`
>
> The corresponding experiments are in progress. We will include additional attack methods for comparison. However, the current attack baseline we adopted already incorporates the idea of regeneration. The regeneration (rinse) method mentioned by the reviewer is conceptually equivalent to WmAttacker in our comparison, with the main difference being the number of execution iterations.
>
> Furthermore, according to the experimental results reported in WAVES, rinse introduces substantial modifications to the image, leading to a significant reduction in visual quality. To ensure a fair comparison, we used single-step regeneration in our experiments to avoid biased results caused by overly aggressive modifications.
>
> `The number of methods evaluated is not enough` and `the limitation to Tree-Ring`
>
> We are still conducting additional experiments. We will include additional algorithms to further enhance the comprehensiveness and objectivity of the experiments. We agree with the reviewer that “the class of diffusion-based watermark is a defining direction of the field.” which is consistent with the discussion provided in Appendix A.1.2 of our paper. Similar to WmAttacker[1], we also explicitly acknowledge in the manuscript that our method is currently less effective on this category of watermarking algorithms.
>
> However, the primary focus of our work is not in-processing watermarking. Instead, our study centers on the potential risks introduced by the robustness-oriented design of post-process image watermarking frameworks. Furthermore, post-process image watermarking is widely adopted in both industry and academia due to its broad applicability and low deployment cost, and it remains a mainstream paradigm in practical watermarking systems, and thus its security deserves continued investigation.
>
> [1]"Invisible image watermarks are provably removable using generative ai."
>
> `Number of images is not enough (100).`
>
> Our experimental setup follows the configuration used in that paper[2]
>
> [2] "Evading watermark based detection of ai-generated content."
>
> `Algorithm may be slow as it needs to do it per image.`
>
> Our method is fully automated. The reason we process each image individually is that the attack scenario we consider is closer to real-world settings: watermark information is often dynamic (e.g., including timestamps). Moreover, as discussed in Appendix A.3.6 and Sec. 4.2, watermarking models tend to couple watermark information with the content of the carrier image, which makes per-image targeted attacks a more suitable and realistic choice.
>
> The reviewer’s suggestion to “create a general mapping” appears to rely on joint optimization across multiple images. However, this may introduce noise into the extracted leakage information and could have adverse effects on forgery attacks that rely more heavily on the coupling between the watermark information and the image content.
>
> # question
> `More insight on 'training watermarking models with increasingly diverse and severe distortions induces embedding redundancy'.`
>
> Given a fixed watermark capacity $k$ and a visual quality constraint $\Delta_{TV}$, as the noise intensity $\delta_{\eta}$ continues to increase, the watermark embedding strength must be increased in order to still satisfy the robustness constraint. This inevitably leads to a degradation in visual quality. Let the channel capacity be $R$:
> $$R=\frac{1}{2}\log(1+\frac{||I_{wm} - I||^2}{\delta_{\eta}^2})$$
> To reliably transmit $k$ bits within error rate $\mathcal{B}$, it must satisfy:
> $$ R \ge k(1 - \mathcal{B})
>     \Rightarrow ||I_{wm} - I||^2 \ge (2^{2k(1 - \mathcal{B})}-1)\delta_{\eta}^2$$This defines the robustness requirement, and the visual quality constraint is satisfied as:
>     $$||I_{wm} - I|| \le \Delta_{TV}||I||$$
> When the noise $\eta$ increases to: $\sqrt{(2^{2k(1 - \mathcal{B})}-1)}\delta_{\eta} \gt \Delta_{TV}||I||$, the system cannot satisfy both constraints; maintaining robustness requires increasing $\Delta_{TV}$(embedding strength), inevitably degrading visual quality.
>
> `Confused by why outlier features/groups with fewer samples are examples of leakages.`
>
> In the paper, we attempted to use a feature network to extract artifact-related features and visualize them. The observation is intuitive: if the artifacts were present in all features, the carrier image would be overwhelmed by such artifacts, violating the fundamental requirement of watermark invisibility. Therefore, the fact that artifacts appear only in a subset of features is consistent with the characteristics of watermarking mechanisms and further supports the motivation and observations behind our method.
>
> `Fully parameterize the optimization procedure`
>
> See `Algorithm may be slow as it needs to do it per image.`

---

> ### Author Response · Authors · 2025-11-23
>
> We have additionally included three watermarking algorithms that are not the primary focus of our work. For Vine, we used the same COCO dataset for watermark embedding as in the original paper. For Stable Signature and GS, we randomly selected 100 prompts from the COCO dataset to generate watermarked images.
>
> We conducted both removal and forgery attacks on three watermarking algorithms.
>
> | ALGO | SR | PSNR | SSIM |
> | -- | -- | -- |-- |
> | VINE$(FPR=10^{-4})$  |   0.97   | 34.21    | 0.72     |
> | GS$(FPR=10^{-5})$  |   0.81   | 34.13    | 0.72     |
> | Stable Signature$(FPR=10^{-6})$ |   0.34   | 34.77 | 0.74   |
>
> Table 1. Experimental results of the forgery attack
>
>
> | ALGO | SR | PSNR | SSIM |
> | -- | -- | -- |-- |
> | VINE$(FPR=10^{-4})$  |   0.34   | 32.32   | 0.62     |
> | GS$(FPR=10^{-5})$  |   0.32   | 34.42    | 0.74     |
> | Stable Signature$(FPR=10^{-6})$ |   0.68   | 32.57    | 0.74  |
>
> Table 2. Experimental results of the removal attack
>
>
>
> In addition, we added three augmentation methods: rotation, brightness+contrast and cropping.
>
> | ALGO | SR | PSNR | SSIM |
> | -- | -- | -- |-- |
> |PIMoG|0.37|17.782|0.315|
> | HiDDeN  |   0.0   | 21.328   | 0.286     |
> | StegaStamp  |   0.65   | 20.82    | 0.356     |
> | CIN |   0.83   | 22.441    | 0.416  |
> | RivaGan |   0.0   | 21.086    | 0.315  |
>
> Table 3. Experimental results of the rotation(7&deg;)
>
>
>
> | ALGO | SR | PSNR | SSIM |
> | -- | -- | -- |-- |
> |PIMoG|0.0|15.911|0.658|
> | HiDDeN  |  0.02   | 18.826   | 0.593     |
> | StegaStamp  |   0.01   | 19.918    | 0.88     |
> | CIN |   0.0   | 15.931    | 0.656  |
> | RivaGan |   0.02   | 18.914    | 0.757  |
>
> Table 4. Experimental results of the brightness+contrast. This method adjusts the brightness and contrast of an image. It has two parameters a (for contrast) and b(for brightness ), where each pixel value x is transformed to ax+b. (a=1.5,b=0.2)
>
>
>
> | ALGO | SR | PSNR | SSIM |
> | -- | -- | -- |-- |
> |PIMoG|0.53|21.111|0.371|
> | HiDDeN  |   0.0   | 21.426   | 0.278     |
> | StegaStamp  |   0.53   | 20.683    | 0.368     |
> | CIN |   0.63   | 22.368    | 0.406 |
> | RivaGan |   0.0   | 20.824    | 0.298  |
>
> Table 5. Experimental results of the crop(10\%).

---

### Meta-Review · Area_Chair_meFk · 2026-01-07

**Summary:**

This paper presents an attack framework for watermark removal and forgery that operates in a no-box setting using only a single watermarked image. The problem is timely and practically relevant, and reviewers appreciated the unified approach to both removal and forgery attacks.

That said, reviewers raised several concerns that limited their enthusiasm about the paper. The primary issue is that the method's scope is limited to post-processing watermarks, while the framing suggests broader applicability. The author's exclusive focus on post-processing watermarks needs a better motivation. Furthermore, the authors acknowledge their approach does not perform well on in-processing methods like Tree-Ring, which limits the generality of the contribution given the growing importance of such methods. Reviewers also felt the evaluation could be more comprehensive, suggesting additional baselines and attack methods. The original submission computed improvements against weak baselines, and when measured against the second-best method, the gains are more modest. Some experimental details were also missing or unclear, though the authors were responsive in clarifying these during the rebuttal. Notably, there was no strong champion for this paper.

**Reviewer Concerns:**

The authors addressed concerns about missing experimental parameters and committed to revising the overreaching claims. They also conducted additional experiments on requested baselines during the rebuttal, which was appreciated.

The main outstanding issue is the scope limitation. While post-processing watermarks remain practically important, the limited effectiveness on in-processing methods constrains the overall contribution. The supplementary experiments were helpful but would benefit from integration into the main part of a revised paper. When checking the revised paper, I could not find the additional watermarking baselines (VINE, Gaussian Shading, Stable Signature) that were shown in the rebuttal and the additional attack methods (rotation, brightness+contrast, cropping) are also not integrated into the paper.

**Reviewer Scores:**

Reviewer QP2W (score 6) engaged constructively and would likely maintain their score since they were already borderline in their initial review. The reviewer acknowledged the authors' responsiveness but noted that core issues remain. Reviewers DC2f (score 4) and PmLJ (score 2) raised many concerns most of which were addressed by the authors, but neither reviewer participated in the discussion. As noted by the authors, PmLJ's review contained factual errors but even after excluding it, the remaining reviews do not champion acceptance.

---

### Decision · Program_Chairs · 2026-01-26

Reject